# STABILIZING MoE REINFORCEMENT LEARNING BY ALIGNING TRAINING AND INFERENCE ROUTERS

## ABSTRACT

Reinforcement learning (RL) has emerged as a crucial approach for enhancing the capabilities of large language models. However, in Mixture-of-Experts (MoE) models, the routing mechanism often introduces instability, even leading to catastrophic RL training collapse. We analyze the training-inference consistency of MoE models and identify a notable discrepancy in routing behaviors between the two phases. Moreover, even under identical conditions, the routing framework can yield divergent expert selections across repeated forward passes. To address this foundational inconsistency, we propose **Rollout Routing Replay (R3)**, a method that records routing distributions from the inference engine and replays them during training. R3 significantly reduces training-inference policy KL divergence and mitigates extreme discrepancies without compromising training speed. Extensive experiments on various settings confirm that R3 succeeds in stabilizing RL training, preventing collapse and outperforming methods such as GSPO and TIS. We believe this work can offer a new solution for stabilizing RL in MoE models.

## 1 INTRODUCTION

Reinforcement learning (RL) has become a cornerstone in the post-training of large language models (LLMs) (Ouyang et al., 2022; OpenAI, 2024; Guo et al., 2025). By leveraging large-scale RL, LLMs acquire the advanced capabilities necessary to tackle complex problems, including competition-level mathematics (Guo et al., 2025) and practical code agent tasks (Luo et al., 2025a), through more profound and extended reasoning.

A critical challenge in LLM RL is balancing efficiency with stability, where the latter is paramount for reliable performance. Modern RL frameworks typically employ distinct engines for inference and training phases (e.g., SGLang (Zheng et al., 2024) for rollout and Megatron (Shoeybi et al., 2019) for training). This architectural separation can lead to divergent token probabilities, potentially causing catastrophic RL collapse (He & Lab, 2025). To mitigate this discrepancy, Yao et al. (2025) incorporate importance sampling mechanism in policy updating, while He & Lab (2025) introduce specialized compute kernels to reduce nondeterminism during LLM Inference. However, in practice, existing approaches do not fully resolve the intensified off-policy issue that arises during RL training on Mixture-of-Experts (MoE) models.

In this work, we identify routing distribution as a pivotal contributing to the instability of MoE RL. Within MoE models, the router dynamically selects and activates a subset of experts for each input token. The varied routing decisions result in greater policy discrepancies between training and inference in MoE models compared to their dense counterparts. Rather than resorting to workarounds like discarding data with excessive discrepancy (Zhao et al., 2025), we propose to tackle this instability by addressing its root cause: the routing distribution itself.

Specifically, we propose **Rollout Routing Replay (R3)**, a simple yet effective method for stabilizing RL training of MoE models. R3 works by capturing the routing distributions from the inference engine during sequence generation and replaying them directly into the training engine. This process significantly narrows the gap between training and inference, marked by a substantial reduction in KL divergence of logits produced by the different engines. As a result, the number of tokens with significant probability discrepancies between the two phases is reduced by approximately an order of magnitude.

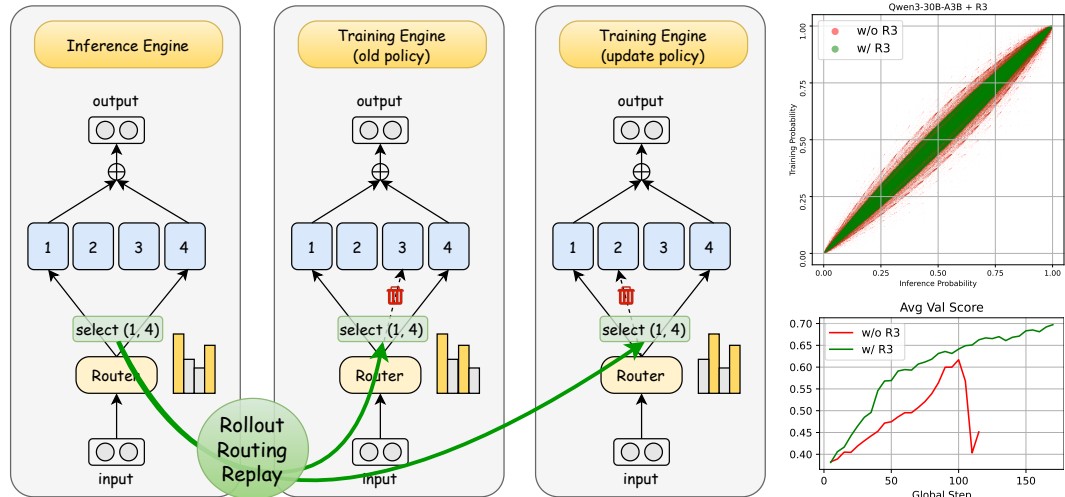

Figure 1: Left: Illustration of the Rollout Routing Replay (R3). Top right: Training and inference discrepancies before and after applying R3. Bottom right: Reinforcement learning training performance before and after applying R3.

On realistic RLVR tasks with MoE models, R3 demonstrates significant superiority in training stability, and performance. Our method consistently shows marked improvements in both efficiency and overall performance when compared against existing approaches designed to stabilize RL training. Furthermore, its applicability to both on-policy and mini-batch style off-policy RL scenarios underscores the robustness of our approach.

Our main contributions are as follows:

1. We systematically identify and analyze routing distribution discrepancies between training and inference in MoE models, highlighting their role in training instability.

2. We proposed Rollout Routing Replay, which reuses inference-time routing distributions inside the training engine to align routing behavior between training and inference.

3. We apply R3 in multiple RL settings (multi-/single-mini-step and Base/SFT models) for MoE reinforcement learning and show that R3 outperforms GSPO and TIS in terms of stability and overall performance.

## 2 PRELIMINARIES

### 2.1 NONDETERMINISM IN MODERN ML SYSTEMS

Modern machine learning systems often exhibit nondeterministic behavior due to subtle numerical and implementation-level factors.

**Floating-point addition is non-associative:** Floating-point addition is non-associative, meaning that the result depends on the order of summation. Limited precision and rounding errors can cause discrepancies when the execution order changes, a problem that becomes especially pronounced in iterative computations and parallel reductions. In concurrent environments, the order of updates to shared memory locations is also influenced by hardware scheduling rather than a deterministic rule. Consequently, operations such as `torch.scatter_add` may yield different outcomes across runs, even when inputs and parameters are identical.

**Violation of batch invariance:** As stated in (He & Lab, 2025), in dynamic batching systems, samples may shift positions across inference iterations due to variable sequence lengths or asynchronous data scheduling. This change in batch composition can modify intermediate model states and influence probability outputs. Similar effects may occur in training pipelines where identical inputs processed in different batch permutations lead to observable variation.

**Variation in operator and kernel selection:** Nondeterminism also arises from Variation in operator and kernel selection across runtime configurations. Frameworks may call backend kernels depending on hardware, batch size, or data type. Although these kernels are mathematically equivalent, they often differ in execution order or numerical stability properties. Consequently, model outputs may diverge slightly across executions, even under nominally identical settings.

**Amplification Effect of the MoE Router:** In deep models, small numerical differences can add up across layers. Each layer changes its input slightly, so tiny floating-point variations can gradually alter internal activations and shift the final outputs. However, MoE models make this effect even stronger because of the router's discontinuous top-K selection. Small changes in routing scores can switch which experts are used, causing the model to follow completely different computation paths. This discrete change can greatly increase the impact of minor numerical differences, leading to larger variations in outputs and potentially affecting processes like reinforcement learning. The proposed R3 method is specifically designed to address this problem.

## 2.2 POLICY OPTIMIZATION ALGORITHM IN REINFORCEMENT LEARNING OF LLM

**Notation** We consider an autoregressive language model, parameterized by $\theta$, as a policy $\pi_\theta$ that generates a response $y$ from a query $x \in \mathcal{D}$. The likelihood of the sequence is given by the factorization $\pi_\theta(y|x) = \prod_{t=1}^{|y|} \pi_\theta(y_t|x, y_{<t})$, where $|y|$ is the sequence length. $\pi_{\text{infer}}$ and $\pi_{\text{train}}$ denote the policy as it operates within the inference and training engines, respectively.

**Proximal Policy Optimization (PPO)** (Schulman et al., 2017) is a cornerstone algorithm for policy optimization in reinforcement learning. For a given query $x$, PPO updates the policy $\pi_\theta$ by maximizing:

$$\mathcal{J}_{\text{PPO}}(\theta) = \mathbb{E}_{x \sim \mathcal{D}, y \sim \pi_{\text{infer}}(\theta_{\text{old}})(\cdot|x)} \left[ \frac{1}{|y|} \sum_{t=1}^{|y|} \min \left( w_t(\theta)\hat{A}_t, \text{clip}(w_t(\theta), 1-\varepsilon, 1+\varepsilon)\hat{A}_t \right) \right], \quad (1)$$

The importance sampling ratio $w_t(\theta)$ for the token $y_t$ in the sequence $y$ is defined as:

$$w_t(\theta) = \frac{\pi_{\text{train}}(\theta)(y_t|x, y_{<t})}{\pi_{\text{train}}(\theta_{\text{old}})(y_t|x, y_{<t})}. \quad (1)$$

The advantage $\hat{A}_t$ for token $y_t$ is typically estimated by a separate value model, and $\varepsilon$ is the clipping range for the importance ratio. For brevity, we omit the KL regularization term.

A critical inconsistency arises from the common practice of using separate engines for rollout and training, where data is sampled via an inference policy ($\pi_{\text{infer}}$) but the loss is computed using a training policy ($\pi_{\text{train}}$), as shown in Equation 1. This policy mismatch drives training instability in reinforcement learning, an issue that, within MoE models, we identify as stemming primarily from router inconsistency. The proposed solution alleviates this fundamental issue, making it broadly applicable, orthogonal to, and compatible with recent policy optimization frameworks like GRPO (Shao et al., 2024), GSPO (Zheng et al., 2025) and DAPO (Yu et al., 2025).

## 3 TRAINING-INFERENCE DISCREPANCIES

Training-inference discrepancies in RL frameworks frequently lead to unstable training and model collapse. In this section, we demonstrate that this policy mismatch is significantly amplified in MoE models, primarily stemming from inconsistent routing distributions Furthermore, we observe that even multiple runs of the same training framework can produce divergent token probabilities, further contributing to RL training instability.

### 3.1 POLICY DISCREPANCIES BETWEEN TRAINING AND INFERENCE IN MoE MODELS

Here we employ an MoE model, Qwen3-30B-A3B (Yang et al., 2025), for our experiments to analyze the policy discrepancy between the training and inference engines. First, we use the SGLang inference engine to generate responses for 2,048 mathematics problems, saving the token probabilities for each generated token. This process yields about 20 million response tokens. Then these

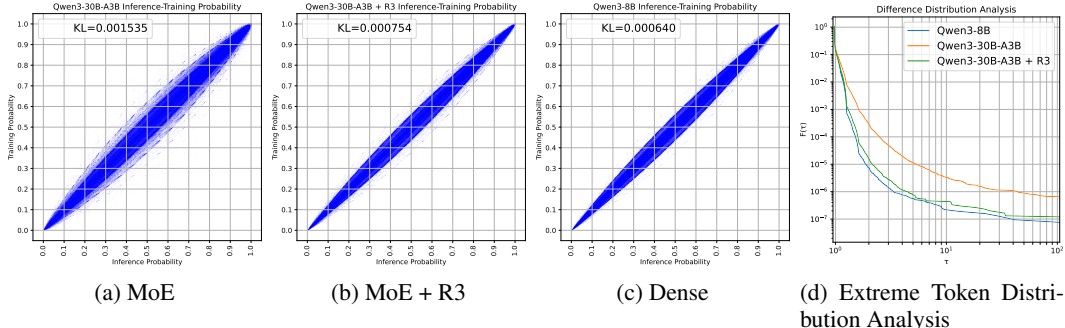

(a) MoE      (b) MoE + R3      (c) Dense      (d) Extreme Token Distribution Analysis

Figure 2: (a): Illustration of the training-inference discrepancy in the MoE model. (b): Illustration of the training-inference discrepancy in the MoE+R3 model. (c): Illustration of the training-inference discrepancy in the Dense model. (d): Extreme Token Distribution Function, calculated based on Equation 3.

responses are passed through Megatron to obtain the corresponding probabilities assigned by the training engine. We use several metrics to quantify the divergence between these two probability distributions.

**KL Divergence Estimation** Let $T$ be the set of all response tokens, we estimate the KL-divergence (Schulman, 2020) between training and inference probability at token level:

$$\mathcal{D}_{\mathrm{KL}}(\pi_{\mathrm{train}}(\theta); \pi_{\mathrm{infer}}(\theta)) \approx \frac{1}{|T|} \sum_{t \in T} \Big[ \frac{\pi_{\mathrm{train}}(\theta)(t)}{\pi_{\mathrm{infer}}(\theta)(t)} - 1 - \log \frac{\pi_{\mathrm{train}}(\theta)(t)}{\pi_{\mathrm{infer}}(\theta)(t)} \Big], \tag{2}$$

Our calculations show that the estimated KL divergence of Qwen3-30B-A3B (MoE) is $1.535 \times 10^{-3}$, while that of Qwen3-8B (Dense baseline) is $6.4 \times 10^{-4}$.

**Visualization** To visualize the training-inference discrepancies of the MoE model, we randomly sample 10 million response tokens and plot a scatter diagram with SGLang probabilities on the x-axis and Megatron probabilities on the y-axis. The degree of concentration around the $y = x$ line indicates the degree of consistency. As shown in Figure 2a and 2c, compared to Qwen3-8B, Qwen3-30B-A3B exhibits a much wider scatter band, revealing a larger training-inference discrepancy.

**Extreme Token Distribution Analysis** To quantify the discrepancy between the model's behavior during training and inference, we introduce the Extreme Token Distribution Function, $F(\tau)$, defined as:

$$F(\tau) = \frac{1}{|T|} \sum_{t \in T} \mathbf{I} \Big[ \max \Big( \frac{\pi_{\mathrm{train}}(\theta)(t)}{\pi_{\mathrm{infer}}(\theta)(t)}, \frac{\pi_{\mathrm{infer}}(\theta)(t)}{\pi_{\mathrm{train}}(\theta)(t)} \Big) > \tau \Big]. \tag{3}$$

This function measures the proportion of *extreme tokens*—those for which the probability ratio between the training distribution $\pi_{\mathrm{train}}(\theta)$ and the inference distribution $\pi_{\mathrm{infer}}(\theta)$ surpasses a threshold $\tau$. Figure 2d plots this function $F(\tau)$ against the threshold $\tau$. The plot reveals that for $\tau > 2$, the Qwen3-30B-A3B model exhibits a fraction of extreme tokens an order of magnitude larger than that of the Qwen3-8B model. This significant gap indicates a substantially higher degree of training-inference variability in the MoE model.

### 3.2 ROUTING DISCREPANCIES BETWEEN TRAINING AND INFERENCE IN MoE MODELS

From the perspective of functional continuity, the key difference between MoE and Dense models lies in the non-continuity introduced by routing. In MoE models, small perturbations in the router input can lead to entirely different experts being selected, causing large changes in layer outputs. Dense models, lacking an explicit expert selection process, do not exhibit this phenomenon.

Based on this, we further analyze router distribution discrepancies between training and inference in MoE models. We use SGLang (Zheng et al., 2024) to generate responses for 2048 mathematical problems with Qwen3-30B-A3B (Yang et al., 2025). For each response, we collect the routing

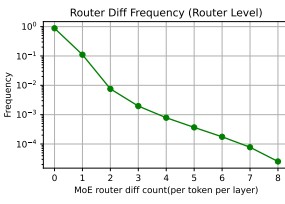
(a) Router-level Difference

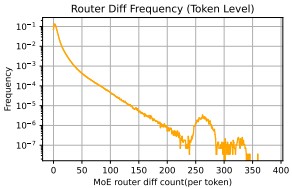
(b) Token-level Difference

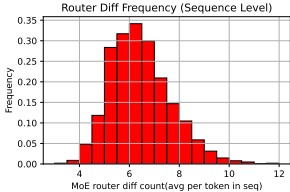
(c) Sequence-level Difference

Figure 3: Router discrepancy analysis

distribution of all tokens (including input tokens) from the inference engine. Then, we feed these sequences into the Megatron engine for forward propagation. This process yields the routing distribution observed from training engine. We compare these two sets of routing information at different levels:

**Router-level Comparison**: For each token and each MoE layer, we count the number of differing expert choices made by the MoE Router and calculate the frequency of these differences. Figure 3a illustrates the results. It is observed that approximately 10% of the routers select different experts during training compared to inference.

**Token-level Comparison**: For each token, we count the number of differing expert choices made by the MoE Router across all layers. We also determine the frequency of these occurrences. Figure 3b presents these findings. It shows that 94% of tokens select a different expert in at least one layer during the forward pass.

**Sequence-level Comparison**: For a sequence, we compute the router distribution difference of each token, then average over tokens to obtain the mean difference per sequence and plot a histogram 3c. Results show that the mean difference per token is approximately 6 routers.

These findings demonstrate that during training and inference, MoE models exhibit router distribution discrepancies. In 4 we show empirically that routing discrepancies is the main contributor to the additional training-inference discrepancies of MoE compared to Dense models.

### 3.3 VARIATION IN REPEATED FORWARD PASSES OF THE SAME SEQUENCE WITHIN THE SAME FRAMEWORK

We conduct two forward passes of the same sequence set under the Megatron framework and obtain two probability distributions. Following the procedure in Section 3.1, we compute the KL divergence between these distributions and plot the results (scatter plot shown in Figure 4, KL divergence = $8.4 \times 10^{(-4)}$).

The results show that, for MoE models in the Megatron engine, even when the input sequence is identical, the final output probabilities from two forward passes may differ. In a reinforcement learning setting, this variation adds noise to the computation of the old policy $\pi_{\text{train}}(\theta_{\text{old}})$. Such noise makes the importance sampling ratio unreliable, which can destabilize and even break the reinforcement learning process.

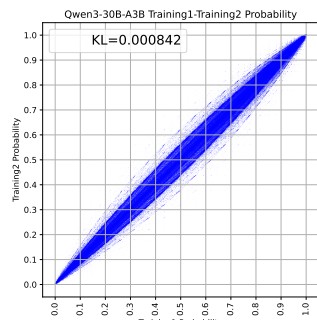

Figure 4: Probabilities obtained by performing forward propagation twice using the Megatron

## 4 ROLLOUT ROUTING REPLAY

This section provides a detailed description of the implementation of Rollout Routing Replay (R3), its caching support in multi-turn dialogue, and an analysis of its effects on training-inference discrepancies.

## 4.1 Implementation

We first describe the conventional forward pass of a MoE layer within a training framework. Consider a sequence $s$ at the $t$-th token and the MoE layer in the $l$-th Transformer block. Let the input to this layer during training be $\mathbf{x}_{\text{train}}$. The router logits are calculated by:

$$\mathbf{s}_{\text{train}} = \mathbf{x}_{\text{train}} \mathbf{W}_r, \tag{4}$$

where $\mathbf{W}_r$ denotes the router's linear weight matrix. Let the number of experts be $M$ and the number of experts to select be $K$. During training, the router selects the top-$K$ experts based on the logits. This is represented by a binary mask:

$$\mathbf{I}_{\text{train}} = \text{TopKMask}(\mathbf{s}_{\text{train}}, K), \tag{5}$$

where $\mathbf{I}_{\text{train}} \in \{0,1\}^M$ and $\sum_i I_{\text{train},i} = K$. The gating weights are then produced by applying a softmax over the logits of the selected experts:

$$g_{\text{train},i} = \frac{I_{\text{train},i} \, \exp(s_{\text{train},i})}{\sum_{j=1}^{M} I_{\text{train},j} \, \exp(s_{\text{train},j})} \quad \text{for } i = 1, \dots, M. \tag{6}$$

Finally, the MoE layer's output is computed as a weighted sum of the expert outputs:

$$\mathbf{y}_{\text{train}} = \sum_{i=1}^{M} g_{\text{train},i} \, \mathcal{E}_i(\mathbf{x}_{\text{train}}), \tag{7}$$

where $\mathcal{E}_i(\cdot)$ represents the $i$-th expert network.

Now we introduce the Rollout Routing Replay. Assume that during the inference stage, the input to the MoE layer is $\mathbf{x}_{\text{infer}}$. The router then computes the inference logits $\mathbf{s}_{\text{infer}} = \mathbf{x}_{\text{infer}} \mathbf{W}_r$, from which the routing mask $\mathbf{I}_{\text{infer}} = \text{TopKMask}(\mathbf{s}_{\text{infer}}, K)$ is obtained. The main idea of Rollout Routing Replay is to reuse the inference routing mask $\mathbf{I}_{\text{infer}}$ during the training forward pass while still applying the softmax to the training logits to preserve gradient flow. Specifically, along this replay path, the "replay" gating weights $g_{\text{replay}}$ are computed as:

$$g_{\text{replay},i} = \frac{I_{\text{infer},i} \, \exp(s_{\text{train},i})}{\sum_{j=1}^{M} I_{\text{infer},j} \, \exp(s_{\text{train},j})} \quad \text{for } i = 1, \dots, M. \tag{8}$$

These replay weights are then used to combine the training experts' outputs, producing the replay output $\mathbf{y}_{\text{replay}}$:

$$\mathbf{y}_{\text{replay}} = \sum_{i=1}^{M} g_{\text{replay},i} \, \mathcal{E}_i(\mathbf{x}_{\text{train}}). \tag{9}$$

This design serves two main purposes: (a) **Aligning training and inference:** Using $\mathbf{I}_{\text{infer}}$ ensures that the experts used during training replay match those selected during inference, eliminating mismatch in expert selection. (b) **Preserving the gradient data flow:** By replaying the mask, we can precisely optimize the experts used when inferring this sequence, avoiding the optimization of unused experts, which may introduce gradient noise.

## 4.2 Router Mask Caching and Multi-Turn Dialogue Support

Many inference engines use a KVCache prefix caching strategy (Zheng et al., 2024; Kwon et al., 2023) to prevent redundant prefill computations on previously seen context, which significantly reduces the total computation in multiple-turn interactions. We observe that the cached router masks share a similar property: for the same prefix tokens, the MoE router should yield identical results. Therefore, routing masks from inference engines $\mathbf{I}_{\text{infer}}$ can be cached along with the prefix KVcache. Concretely, for each layer and token prefix, the corresponding routing masks are stored with KVCache. When the same prefix occurs and hits the cache, the masks can be reused, eliminating the need for recomputation. This allows Rollout Routing Replay to integrate seamlessly with prefix caching mechanisms.

Caching routing masks is especially beneficial in agent scenarios. Many agent tasks, like software engineering (Jimenez et al., 2024) and web browsing (Wei et al., 2025), involve multiple turns

of interactions between autoregressive generation and tool calling. To improve efficiency, these processes directly reuse the KVCache from previous turns, so they do not have to regenerate data that's already been computed. Routing mask caching enables R3 to remain efficient in RL agent tasks without re-prefilling to generate the routing masks, which is crucial for training large-scale, advanced MoE models.

## 4.3 EMPIRICAL ANALYSIS OF R3 ON TRAINING-INFERENCE DISCREPANCIES

To evaluate the effectiveness of R3 in reducing training-inference discrepancies, we repeat the procedure described in Section 3.1 with Qwen3-30B-A3B model. In this process, we cache the routing distributions obtained during inference on SGLang and replay them within the Megatron framework. After obtaining the inference engine probability and training engine probability, using the equation 2, we estimate the KL divergence between training and inference. Results shows that after applying R3, the KL divergence between training and inference decrease from $1.5 \times 10^{-3}$ to $7.5 \times 10^{-4}$, which is near the $6.4 \times 10^{-4}$ observed for the dense model. This intuitively indicates a reduction in the training-inference discrepancy. We also drew the cumulative distribution plot of the ratio of training-inference discrepancies in Fig. 2d with R3. The plot indicates that, for the MoE model, applying R3 reduces by an order of magnitude the frequency of tokens with large training-inference discrepancies.

## 5 EXPERIMENTS

In this section, we evaluate the performance improvement of our R3 method for reinforcement learning and compare it with other baseline methods.

### 5.1 SETTING

**Task and Dataset** We choose mathematical reasoning as the target task for training. For the training dataset, we collect and filter problems from many open-source datasets, including BigMath (Albalak et al., 2025), ORZ (Hu et al., 2025), and others, yielding approximately 100,000 verifiable math problems. For the evaluation dataset, we adopt AIME24, AIME25, AMC23, and MATH500 (level 5) as our benchmark datasets. We report Avg@32 for AIME24 and AIME25, Avg@16 for AMC23, and Avg@4 for MATH500 (level 5). During training, some models may experience performance degradation or even collapse in later stages. To ensure a fair evaluation, we measure model performance every 5 global steps throughout a single training run. We then report the highest observed performance along with the corresponding training step at which it occurs.

**Models** We select two models for our experiments: (a) Qwen3-30B-A3B-Base (Yang et al., 2025) (b) Qwen3-30B-A3B-SFT, which is fine-tuned from Qwen3-30B-A3B-Base on our general instruction-following dataset.

**Baseline Methods** We consider the following baseline optimization methods for comparison: (a) **GRPO** (Shao et al., 2024), additionally applies the *Clip Higher* technique from DAPO (Yu et al., 2025), with parameters $\epsilon_{low} = 0.2$ and $\epsilon_{high} = 0.27$; (b) **TIS** (Yao et al., 2025), uses an upper clipping threshold $C = 2$; (c) **GSPO** (Zheng et al., 2025), employs sequence-level importance sampling, with parameters $\epsilon_{low} = 3 \times 10^{-4}$ and $\epsilon_{high} = 4 \times 10^{-4}$. Since our R3 method is orthogonal to optimizers such as GSPO or TIS, we also evaluate various combinations of these techniques.

**Multiple Mini Steps vs. Single Mini Step** In PPO-like algorithms, a batch of samples collected in one global step is typically divided into multiple mini steps so that the policy can be updated several times. However, previous research has shown that applying a strictly on-policy strategy with only a single mini step can yield better performance. We investigate both settings. For multiple mini steps, we set the mini step count to 8, meaning each PPO mini step trains $2048/8 = 256$ samples with 8 optimizer updates and a learning rate of $1 \times 10^{-6}$. In the multiple mini-step scenario, for R3 method we replayed the routing both when recomputing the old policy and updating policy. For the single mini step scenario, we set the mini step count to 1, meaning all 2048 samples are updated once. Due to fewer updates, we increase the learning rate to $3 \times 10^{-6}$. We don't recompute old policy in this scenario.

| | Best Metric(Best Global Step) | | | | | |
|---|---|---|---|---|---|---|
| Method | AIME24($\uparrow$) | AIME25($\uparrow$) | AMC23($\uparrow$) | MATH500 Lv5($\uparrow$) | Avg($\uparrow$) | Crash Step |
| **Qwen3-30B-A3B-SFT, mini_step=8, max_global_step=180** | | | | | | |
| GRPO | 32.81(65) | 20.73(90) | 74.84(60) | 71.83(100) | 48.84(100) | 120 |
| GSPO | 55.52(165) | 38.23(160) | 90.16(125) | 86.38(125) | 66.76(125) | - |
| GRPO+R3 | 57.92(180) | 38.02(155) | 90.16(155) | **88.62**(170) | 68.05(180) | - |
| GSPO+R3 | **58.44**(160) | **39.17**(160) | **92.50**(165) | 87.87(165) | **69.00**(165) | - |
| **Qwen3-30B-A3B-SFT, mini_step=1, max_global_step=160** | | | | | | |
| GRPO | 49.06(45) | 32.08(50) | 86.41(55) | 83.77(55) | 62.23(55) | 60 |
| GRPO+TIS | 54.90(85) | 36.67(90) | 88.59(90) | 85.63(85) | 66.24(90) | 105 |
| GRPO+R3 | **62.60**(160) | **41.67**(160) | **93.44**(150) | **89.93**(155) | **71.82**(160) | - |
| GRPO+TIS+R3 | 57.40(135) | 39.27(135) | 89.53(115) | 88.62(130) | 67.47(135) | - |
| **Qwen3-30B-A3B-Base, mini_step=1, max_global_step=180** | | | | | | |
| GRPO | 50.63(100) | 32.60(100) | 83.13(100) | 80.78(90) | 61.69(100) | 105 |
| GRPO+TIS | 56.77(170) | 40.10(170) | 92.50(165) | 89.37(180) | 69.22(170) | - |
| GRPO+R3 | **60.94**(180) | **41.35**(180) | 92.81(175) | **89.74**(175) | **70.73**(180) | - |
| GRPO+TIS+R3 | 56.67(170) | 40.42(170) | **93.12**(175) | 88.99(165) | 69.17(175) | - |

Table 1: Main evaluation results. This table presents the best scores (with corresponding global step in parentheses) achieved by various methods on evaluation benchmarks under three training configurations . The table also reports the average score (Avg) and the step at which training collapse occurred (Crash Step).

**Other Setting** We implement R3 using the VeRL (Sheng et al., 2024) framework, use Megatron (Shoeybi et al., 2019) for training, and SGLang (Zheng et al., 2024) for inference. We set the batch size to 256 and $n = 8$, totaling 2048 samples per round. The maximum prompt length is 2048, and the maximum generation length is 30720. We adopt the Dynamic Sampling strategy from Yu et al. (2025), retaining only partially correct samples during generation until a sufficient batch size is accumulated for training. No auxiliary loss for expert balancing is introduced.

## 5.2 EXPERIMENTAL RESULTS

The main evaluation results are shown in Table 1. More detailed evaluation results and training logs, including per-evaluation scores as well as training metrics such as entropy, average reward, gradient norm, and training-inference KL divergence, are provided in the appendix A and A

**Overall Performance.** R3 achieves better results across different scenarios. In the multi mini-step setting, GRPO+R3 outperforms GSPO by 1.29. Furthermore, combining R3 with GSPO further improves performance by 0.95 points. In the single mini-step setting, R3 outperforms TIS by 5.58 on SFT model and 1.51 on base model. However, combining R3 with TIS does not yield clear gains and may even degrade performance; for instance, in the single mini-step on SFT model, TIS+R3 scores 3.29 points lower than R3 alone. Since R3 already significantly reduces the policy discrepancy between training and inference, the additional correction from TIS offers negligible benefit (Yao et al., 2025).

**Training Stability.** In the single mini-step setting, three reinforcement learning processes without R3 collapsed during training. To investigate the causes of these collapses, we plotted the estimated training–inference KL divergence and the extreme token distribution function $F(\tau = 2)$ (eq. 3) at each training step(see Fig. 5). We observed that, in each training process, both the KL divergence and $F(\tau = 2)$ values increased over training process. Moreover, collapsed training runs were almost always accompanied by abnormally high KL and $F(\tau = 2)$ values. For instance, in the SFT model trained using GRPO under the single mini-step setting, after 60 global steps, the value of $F(\tau = 2)$ exceeded 0.1. This indicates that for 10% of the tokens, the probability under the training framework differed from that under the inference framework by more than a factor of 2, demonstrating severe training–inference inconsistency. In contrast, for training processes using R3, the value of $F(\tau = 2)$ remained below $10^{-4}$. By aligning the training and inference distributions, R3 effectively stabilized reinforcement learning for MoE models.

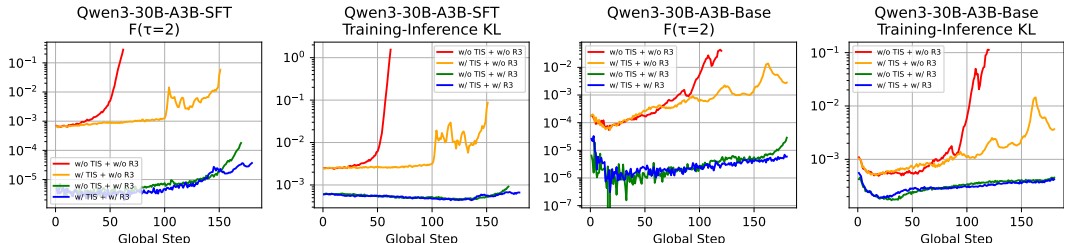

Figure 5: Analysis of training–inference collapse. The plot shows the estimated training–inference KL divergence and the extreme token distribution function $F(\tau = 2)$ (Eq. 3) at each training step.

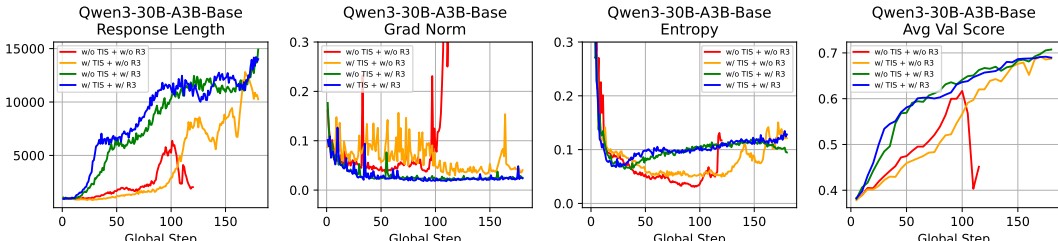

Figure 6: Training dynamics of Qwen3-30B-A3B-Base, including response length, gradient norm, entropy, and average validation score throughout the training process

**Optimization and Generation Behavior.** During training, R3 also enhances optimization stability, exploration behavior, and generation dynamics. We plotted the sequence length, gradient norm, generation entropy, and evaluation score throughout training for the single mini-step + base model group (see Fig. 6). It shows that R3 has smaller gradient norms, a smoother sequence growth pattern, and more stable entropy. (a) **Sequence Length:** With R3, the generated sequence length rises rapidly at the beginning of training, indicating that R3 can quickly capture the correct optimization direction. By contrast, the other two training processes increase only slowly after step 80 and show more pronounced fluctuations. (b) **Gradient Norm:** R3 maintains consistently lower gradient norms, indicating a more stable optimization process. (c) **Generation Entropy:** With R3, entropy begins to increase steadily after about 25 step, suggesting that the model starts exploring better strategies earlier. Without R3, entropy increases much later and fluctuates heavily.

### 5.3 END-TO-END PERFORMANCE TEST

We run end-to-end performance tests on Qwen3-30B-A3B with and without R3. We set batch size to 256, $n = 8$, and maximum generation length to 30720, and run 10 steps of reinforcement learning. The results show that using R3 adds only **3.45%** overhead, which proves that R3 is practical. We describe our performance test process and results in detail in Appendix F, and analyze the extra memory usage of R3 in Appendix G.

## 6 RELATED WORKS

### 6.1 MIXTURE OF EXPERTS

The Mixture-of-Experts (MoE) architecture, originating as an ensemble method to route inputs to specialized subnetworks (Jacobs et al., 1991; Shazeer et al., 2017), has become a cornerstone for scaling modern language models. By employing a gating network to sparsely activate only a subset of expert parameters per token, MoE decouples a model's total parameter count from its inference cost, enabling a substantial increase in model capacity. This computational efficiency has driven its adoption in state-of-the-art Transformer models (Jiang et al., 2024; Liu et al., 2024; Team et al., 2025; Yang et al., 2025).

## 6.2 REINFORCEMENT LEARNING WITH VERIFIABLE REWARDS

Reinforcement Learning with verifiable rewards (RLVR) has become a standard method for refining the complex reasoning (Xie et al., 2025), mathematical (Shao et al., 2024; Hu et al., 2025), and code abilities (Luo et al., 2025b) of large language models. Algorithms such as GRPO (Shao et al., 2024) DAPO (Yu et al., 2025) and GSPOZheng et al. (2025) are now widely adopted. This method has already been used to train state-of-the-art reasoning models (Yang et al., 2025; Xia et al., 2025; Chen et al., 2025; Guo et al., 2025; Team et al., 2025).

## 6.3 NONDETERMINISM, FRAMEWORK DISCREPANCIES AND REINFORCEMENT LEARNING

With the rise of RLVR, training–inference discrepancies and non-determinism issues have also drawn widespread attention, largely due to concerns about training stability. (Chen et al., 2025) pointed out the discrepancies problem between training and inference can cause instability problems in reinforcement learning and alleviated it by raising the output layer's precision to FP32. (Yuan et al., 2025) examined how numerical precision affects inference nondeterminism. (He & Lab, 2025) discussed the main reasons why inference can become nondeterministic. (He & Lab, 2025) discussed the main reasons why inference can become nondeterministic. From the perspective of importance sampling, (Yao et al., 2025), (Zhao et al., 2025), and (Liu et al., 2025) proposed the TIS, Icepop, and MIS methods to address the problem. (Qi et al., 2025) pointed out that using FP16 is sufficient to resolve the training-inference discrepancies issue.

## 7 DISCUSSION

### 7.1 R3 V.S. LOSS CORRECTION

Loss correction(GSPO, TIS) only adjusts the loss by applying the importance sampling ratio, but it does not correct the forward routing distribution. Thus, gradients under TIS and GSPO are still distributed to experts that were not used during inference, which means the discrepancies between training and inference remains. In contrast, R3 modifies the forward pass by replaying the exact routing distribution from inference. This ensures that gradients flow exclusively to the experts that were actually responsible for generating the rollout tokens, thereby reducing gradient noise. In short: Loss correction just suppresses the symptoms, whereas R3 corrects the root cause.

### 7.2 FUTURE WORK

Future work can proceed in the following directions: (a).**Infrastructure: Absolute alignment without slowdown** R3 achieves partial alignment on MoE models without causing slowdown. Some existing work has attempted to achieve absolute (bit-level) alignment between training and inference. However, current open-source efforts have only aligned Dense models, not MoE models, and suffer from significant slowdown. Future research should aim to achieve near bit-level alignment between training and inference on all mainstream architectures with minimal or no slowdown. (b).**Algorithm: Understanding the growth of training-inference discrepancies** It is known that the fundamental cause of training-inference discrepancies lies in floating-point operations. However, the reason why these discrepancies grow during the RL process remains unclear. Providing an algorithmic explanation for this growth would deepen our understanding of RL stability.

## 8 CONCLUSION

In this work, we identify training-inference routing discrepancies as the primary source of instability in MoE reinforcement learning. To address this, we propose Rollout Routing Replay (R3), which reuses inference-time routing distributions during training to align expert selection while preserving gradient flow. Experiments across multiple RL settings demonstrate that R3 substantially reduces training-inference divergence, stabilizes training, and consistently outperforms existing methods. Our results demonstrate the importance of aligning training and inference in MoE models and show that R3 provides a practical solution for improving stability.

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

# A    DETAILED EVALUATION RESULTS AND TRAINING METRICS

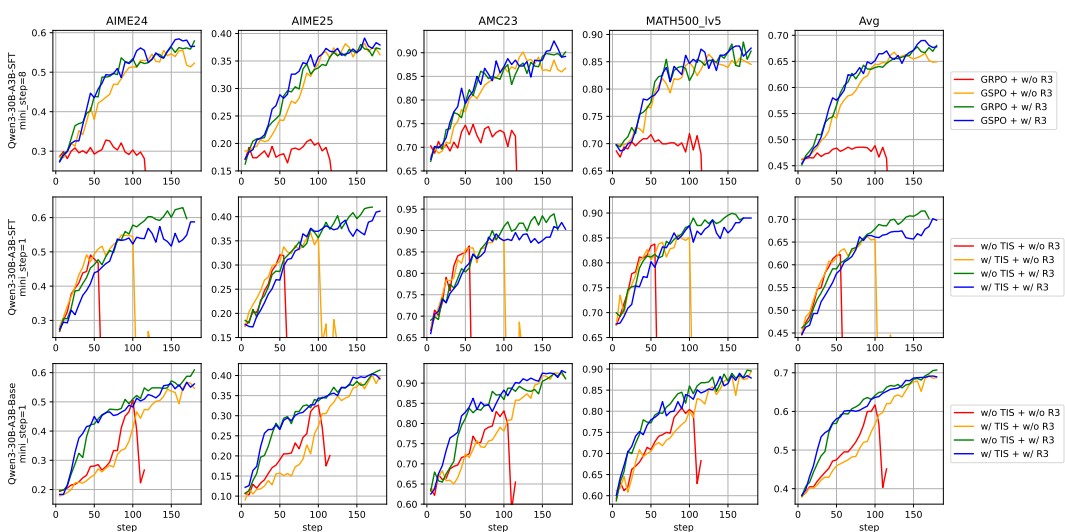

Figure 7: The detailed evaluation results of the experiment in Section 5.

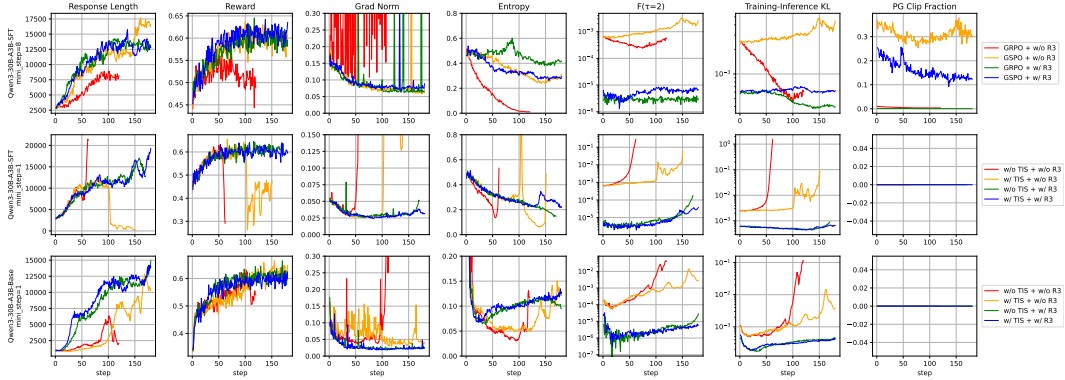

Figure 8: The detailed training metrics of the experiment in Section 5.

# B    OTHER MODEL EXPERIMENTS

## B.1    EXPERIMENTS ON DEEPSEEK-V2-LITE

To evaluate the performance of R3 on other MoE models, we conduct experiments on Deepseek-V2-Lite (DeepSeek-AI et al., 2024). We first fine-tune Deepseek-V2-Lite on the open-source long reasoning dataset Mixture-of-Thoughts [1](Face, 2025; Penedo et al., 2025; Lozhkov et al., 2025; Bercovich et al., 2025), producing the Deepseek-V2-Lite-ReasoningSFT model. We then conduct reinforcement learning experiments on this model.

We compare GRPO and GRPO+R3 with a learning rate of $3 \times 10^{-6}$ when mini_step = 1; all other hyperparameters follow the configuration described in Section 5.1.

The results are presented in Table 3. The experiments show that training runs without R3 eventually collapse (for example, when mini_step = 1, GRPO collapses at training step 100), whereas training

---
[1]https://huggingface.co/datasets/open-r1/Mixture-of-Thoughts

runs with R3 remain stable for up to 250 training steps. These findings demonstrate that the collapse issue occurs not only in Qwen3 MoE but also in other MoE models, and that R3 can likewise ensure stable training across these settings.

| | | | Best Metric(Best Global Step) | | | |
|---|---|---|---|---|---|---|
| Method | AIME24($\uparrow$) | AIME25($\uparrow$) | AMC23($\uparrow$) | MATH500 Lv5($\uparrow$) | Avg($\uparrow$) | Crash Step |
| **Deepseek-V2-Lite-ReasoningSFT, mini_step=1, max_global_step=250** | | | | | | |
| GRPO | 6.87(60) | 9.69(55) | 42.50(50) | 37.50(55) | 23.55(55) | 100 |
| GRPO+R3 | **8.85**(210) | **11.46**(145) | **48.28**(165) | **43.28**(235) | **26.43**(235) | - |

Table 2: Evaluation results of the RL training process of Deepseek-V2-Lite-ReasoningSFT

## B.2 EXPERIMENTS ON QWEN3-8B-BASE

In Section 5.1, we noted that dense models, which lack a router, should naturally exhibit greater stability than MoE models. To verify this point, we conduct reinforcement learning experiments on Qwen3-8B-Base following the same experimental setup as in Section 5.1. Since dense models do not contain a router, R3 cannot be applied, and thus we only perform the baseline experiments.

The results, shown in Table , indicate that the Qwen3-8B-Base model maintains stable training for up to 180 steps. Combined with the findings in Section 5, this demonstrates that dense models inherently possess better stability than MoE models, further confirming that router behavior is a key factor underlying the instability of reinforcement learning in MoE architectures.

| | | | Final Metric | | | |
|---|---|---|---|---|---|---|
| Method | AIME24($\uparrow$) | AIME25($\uparrow$) | AMC23($\uparrow$) | MATH500 Lv5($\uparrow$) | Avg($\uparrow$) | Crash Step |
| **Qwen3-8B-Base, mini_step=1, max_global_step=180** | | | | | | |
| GRPO | 41.35 | 29.06 | 84.53 | 82.65 | 59.40 | - |

Table 3: Evaluation results of the RL training process of Qwen3-8B-Base

## B.3 ADDITIONAL TRAINING-INFERENCE DISCREPANCY

We evaluate $F(\tau) = 2$ and the KL divergence for several models with and without R3, including Qwen3-30B-A3B, Deepseek-V2-Lite, Mixtral-8x7B, and Moonlight-16B-A3B. The results are as follows. "SFT" indicates that the model has been fine-tuned on reasoning data.

| Model & Method | $F(\tau) = 2$ | $KL$ |
|---|---|---|
| Qwen3-30B-A3B | 2.54e-4 | 1.37e-3 |
| Qwen3-30B-A3B + R3 | **5.83e-6** | **7.03e-4** |
| Deepseek-V2-Lite(SFT) | 2.25e-3 | 4.06e-3 |
| Deepseek-V2-Lite(SFT) + R3 | **2.16e-4** | **1.17e-3** |
| Mixtral-8x7B(SFT) | 9.77e-5 | 1.03e-3 |
| Mixtral-8x7B(SFT) + R3 | **1.08e-6** | **4.98e-4** |
| Moonlight-16B-A3B-Instruct | 1.73e-5 | 3.67e-4 |
| Moonlight-16B-A3B-Instruct + R3 | **6.94e-7** | **1.92e-4** |

Table 4: Training-Inference Discrepancy

From the results, it can be seen that R3 can improve training and inference consistency in many types of MoE models. This includes models with different router activations (Qwen3-30B-A3B, DeepSeek-V2-Lite, and Mixtral-8x7B use softmax, while Moonlight-16B-A3B uses sigmoid), different numbers of experts (DeepSeek-V2-Lite and Moonlight-16B-A3B have 64 experts, Qwen3-30B-A3B has 128 experts, and Mixtral has only 8 experts), and whether shared experts are used

## C  FIX ROUTER EXPERIMENTS

We repeated a controlled study on the `Qwen3-30B-A3B-Base` model to investigate whether freezing the router alone prevents training collapse. In this experiment the router parameters were frozen, while all other training hyperparameters and procedures strictly followed the configuration given in Section 5.1. The results are shown in Table 5

We observe that freezing the router does not prevent catastrophic collapse: the model still experiences a pronounced deterioration in performance at approximately training_step = 120. The experiment implies that attributing collapse solely to failures in the router parameters is insufficient. Instead, it suggests that the collapse phenomenon involves degradation of the model parameters as a whole rather than an isolated malfunction of the router.

| | \multicolumn{6}{c}{Best Metric(Best Global Step)} | |
|---|---|---|---|---|---|---|
| Method | AIME24($\uparrow$) | AIME25($\uparrow$) | AMC23($\uparrow$) | MATH500 Lv5($\uparrow$) | Avg($\uparrow$) | Crash Step |
| \multicolumn{7}{c}{**Qwen3-30B-A3B-Base, mini_step=1, max_global_step=180**} |
| GRPO+FixRouter | 49.27(115) | 33.75(115) | 84.38(115) | 82.46(115) | 62.47(115) | 120 |
| GRPO+R3 | **60.94**(180) | **41.35**(180) | **92.81**(175) | **89.74**(175) | **70.73**(180) | - |

Table 5: Evaluation results of the RL training process of fixed router

## D  MULTI-TURN TASK REINFORCEMENT LEARNING EXPERIMENTS

In Section 4.2, we mentioned that the R3 method can be adapted to Multi-Turn tasks. To verify this capability, we conduct reinforcement learning experiments on the software engineering (Jimenez et al., 2024) task using the Qwen3-30B-A3B (Yang et al., 2025) model. We use R2E-Gym-Lite[2] (Jain et al., 2025) as the training dataset and SWE-bench Verified[3] (Jimenez et al., 2024) as the validation dataset. For training, We set the maximum sequence length to $65536$ tokens (including the prompt, model response, and environment observation), the batch size to $64$, the maximum number of interaction steps to $50$, the learning rate to $2 \times 10^{-6}$, and the mini_step to $1$. For evaluation, We report Pass@1 on SWE-bench Verified. We evaluate the model performance every $10$ global steps. We compare the performance of GRPO and GRPO+R3 under these settings. All other hyperparameters follow the configuration described in Section 5.1.

The results are presented in Table 6, and the detailed training metrics are provided in Figure **??**. The training process of GRPO without R3 collapses after around 90 training steps, while GRPO+R3 maintains stable. The final performance of GRPO+R3 reaches a Pass@1 of 38.6, which is 6.8 points higher than that of GRPO. Since we use the Router Mask Caching technique described in Section 4.2, we do not need to re-prefill the prefix to obtain the routing mask, thus the rollout speed is not slowed down. These results confirm that R3 generalizes well to multi-turn reinforcement learning settings.

| \multicolumn{4}{c}{**Qwen3-30B-A3B, mini_step=1, max_global_step=180**} |
|---|---|---|---|
| Method | SWE-bench Verified($\uparrow$) | Best Global Step | Crash Step |
| GRPO | 31.80 | 70 | 90 |
| GRPO+R3 | **38.60** | 160 | - |

Table 6: Evaluation results of RL training on SWE Task with Qwen3-30B-A3B

---

[2]https://huggingface.co/datasets/R2E-Gym/R2E-Gym-Lite
[3]https://huggingface.co/datasets/princeton-nlp/SWE-bench_Verified

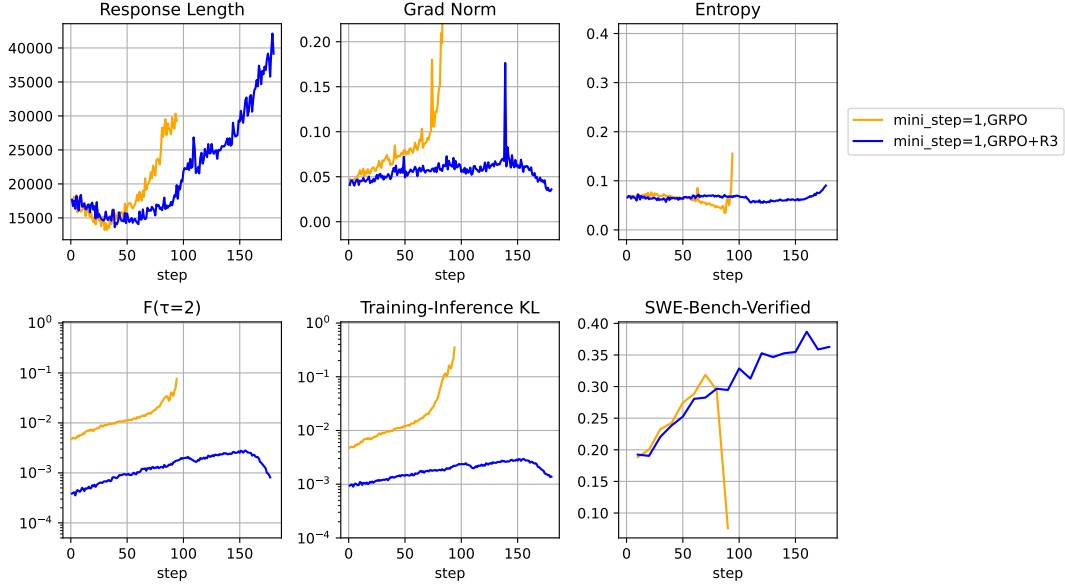

Figure 9: The detailed training and evaluation results of the SWE experiment.

# E    RANDOM SEED EXPERIMENTS

We have conducted three runs under different seeds for GSPO, TIS on Qwen3-30B-A3B-Base model with mini-step=4, learning rate=2e-6. The results are as follows:

| Method | Seed | AIME24(↑) | AIME25(↑) | AMC23(↑) | MATH500 Lv5(↑) | Avg(↑) | Mean±Std |
|---|---|---|---|---|---|---|---|
| | | | Best Metric(Best Global Step) | | | | |
| **Qwen3-30B-A3B-Base, mini_step=4, max_global_step=200** | | | | | | | |
| GSPO | 2025 | 56.35 | 36.56 | 87.13 | 90.47 | 67.63 | |
| GSPO | 2026 | 55.73 | 34.38 | 87.50 | 90.31 | 66.98 | 65.35±3.40 |
| GSPO | 2027 | 45.21 | 35.52 | 80.04 | 85.00 | 61.44 | |
| TIS | 2025 | 47.81 | 33.13 | 84.89 | 86.41 | 63.06 | |
| TIS | 2026 | 49.58 | 32.50 | 84.33 | 86.56 | 63.24 | 63.82±1.16 |
| TIS | 2027 | 51.56 | 35.00 | 85.82 | 88.28 | 65.16 | |
| R3 | 2025 | 52.81 | 37.92 | 84.14 | 88.13 | 65.75 | |
| R3 | 2026 | 57.54 | 39.06 | 88.81 | 91.41 | 69.20 | **67.39±1.73** |
| R3 | 2027 | 56.04 | 35.31 | 88.43 | 89.06 | 67.21 | |

Table 7: Evaluation results of the random seed experiments

From the results, we can see that R3 achieved the best performance. GSPO shows large performance fluctuations between runs and can easily produce bad cases, while TIS performs slightly worse.

## F  END-TO-END PERFORMANCE TEST DETAIL

We conduct end-to-end performance benchmarking to provide a clear comparison between configurations with and without R3. Specifically, the Qwen-30B-A3B model is used, and reinforcement learning training is performed under two settings. The experimental setup is defined as follows: $\text{batch\_size} = 256$, $N = 8$, and $\text{max\_length} = 32K$.

To ensure stable and representative measurements, the following precautions are applied:

- The learning rate is set to 0.0 to disable model parameter updates while still executing the full backward and optimizer steps.

- Dynamic sampling is disabled to eliminate additional variance during measurement.

- Evaluation is disabled to avoid extra computational overhead.

- Each experiment is run for 15 training steps, and only the runtimes of steps 5–14 (10 steps) are used for statistical averaging.

The measured runtime reflects the complete end-to-end execution latency, including inference, forward computation, backward propagation, model optimization, and all communication overhead between the inference and training frameworks. The final benchmarking results are summarized in Table 8.

| Method | # Step | # Token | Runtime (Second) | # Token/s | Relative throughput |
|--------|--------|---------|------------------|-----------|---------------------|
| w/o R3 | 10 | 203,989,395 | 4820 | 42,321 | 100% |
| w/ R3 | 10 | 203,201,992 | 4967 | 40,910 | 103.45% |

Table 8: Throughput analysis result of Rollout Routing Replay

Our findings indicate that, after careful optimization, the inclusion of R3 results in only a 3.45% reduction in end-to-end training throughput. This overhead is minimal and suggests that R3 can be integrated into real-world systems with negligible performance impact. We hope this quantitative analysis addresses the reviewer's request and demonstrates the practicality of deploying R3 in production environments.

## G  ROUTING-MASK CACHING SIZE ANALYSIS

### G.1  ROUTING-MASK CACHE IN INFERENCE FRAMEWORK COMPARED WITH KV CACHE

For the inference framework, the routing-mask cache is stored in CPU memory and is only $\frac{1}{128}$ the size of the KV cache stored in GPU HBM. Routing masks are maintained alongside KV cache entries, but with a significantly smaller memory footprint due to their lower dimensionality and their storage on CPU memory. The datatype of the KV cache is `bfloat16` (2 bytes), and the datatype of the routing mask is `int16` (2 bytes).

Assuming 40 GB of GPU memory allocated for KV cache prefix caching per GPU, only about 2.5 GB of CPU memory per node is required to store routing masks for eight GPUs, which results in negligible deployment overhead. The comparison is summarized in Table 9.

| | KV Cache | Routing Mask Cache |
|--------|----------|--------------------|
| Each Token | DataType $\times$ #Layer $\times$ (KDim + VDim) | DataType $\times$ #Layer $\times$ TopK |
| Each Token in Qwen3-30B-A3B | $2 \times 48 \times (512 + 512)$ | $2 \times 48 \times 8$ |
| 437K Tokens per GPU | 40 GB on GPU HBM | 0.3 GB on CPU Memory |
| 437K $\times$ 8 Tokens per Node (8 GPUs) | 40 GB $\times$ 8 on HBM | 2.5 GB on CPU Memory |

Table 9: Comparison of KV Cache and Routing Mask Cache in the Inference Framework

## G.2 Routing Mask in Training Framework per Global Training Step

For the training framework, the maximum memory footprint of routing masks per global step is determined by the batch configuration. Let the number of prompts be $B$, the number of groups per prompt be $G$, the maximum sequence length be $N$, the number of layers be $L$, and each layer selects $K$ experts. The routing mask stores the expert indices of all selected experts using the `int16` datatype. The maximum routing-mask memory requirement is

$$\text{Mask Size} = 2 \times B \times G \times N \times L \times K \text{ bytes.}$$

For the configuration $B = 256$, $G = 8$, $N = 32768$, $L = 48$, and $K = 8$, the maximum routing-mask memory footprint is approximately 48 GB.

In practice, these masks are distributed across the training infrastructure rather than being stored on a single machine. They are typically stored in CPU memory and only transferred into GPU memory when required during training.

If the router states are distributed across $T$ training nodes—for example, $T = 32$—each machine stores only around 1.5 GB of mask data. For modern AI clusters equipped with high-bandwidth interconnects and large memory capacity, both the storage and communication overhead introduced by such routing masks are well within operational limits.

## H Formal Definition of Router-Discrepancy Metrics

As analyzed in Section 3.2, we have already discussed the qualitative differences in routing behavior. Below, we formally define the meanings of the x-axis and y-axis in Figure 3 using mathematical notation.

Let $D = \{s_1, s_2, \ldots, s_B\}$ denote a batch of sequences. For each sequence $s_i$, let its length be $N_i$, and let the model depth be $L$. Define the router difference during the forward pass as

$d(i, t, l) = \text{difference between router selection during training and inference at layer } l \text{ for token } s_{i,t}.$

**Router-Level:** (Figure 3(a))

$$y = \frac{1}{L \cdot \sum_{i=1}^{B} N_i} \sum_{i=1}^{B} \sum_{t=1}^{N_i} \sum_{l=1}^{L} \mathbf{1}[d(i, t, l) = x].$$

**Token-Level:** (Figure 3(b))

$$y = \frac{1}{\sum_{i=1}^{B} N_i} \sum_{i=1}^{B} \sum_{t=1}^{N_i} \mathbf{1}\left[\sum_{l=1}^{L} d(i, t, l) = x\right].$$

**Sequence-level:** (Figure 3(c))

$$y = \frac{1}{B} \sum_{i=1}^{B} \mathbf{1}\left[x \leq \frac{1}{N_i} \sum_{t=1}^{N_i} \sum_{l=1}^{L} d(i, t, l) < x + 1\right].$$

## I Additional Experiments of Reasoning SFT Model

To evaluate the performance of R3 on reasoning SFT models, we fine-tune Qwen3-30B-A3B-Base (Yang et al., 2025) on the open-source long reasoning dataset Mixture-of-Thoughts (Face, 2025; Penedo et al., 2025; Lozhkov et al., 2025; Bercovich et al., 2025), producing the Qwen3-30B-A3B-ReasoningSFT model. We conduct reinforcement learning experiments on this model. We compare GSPO and GRPO+R3 with a learning rate of $2 \times 10^{-6}$ when mini_step $= 4$, and

compare GRPO and GRPO+R3 with a learning rate of $3 \times 10^{-6}$ when mini_step $= 1$; all other hyperparameters follow the configuration described in Section 5.1.

The results are shown in Table 10, and training and validation metrics are illustrated in Fig 10. The experiments show that training processes without R3 collapse (for mini_step $= 4$, GSPO collapses at training step 90, and for mini_step $= 1$, GRPO collapses at training step 40), while training processes with R3 remain robust. In addition, training with R3 exhibits a more stable pattern of sequence-length growth as well as more stable entropy and gradient norms throughout the training process.

| Method | AIME24($\uparrow$) | AIME25($\uparrow$) | AMC23($\uparrow$) | MATH500 Lv5($\uparrow$) | Avg($\uparrow$) | Crash Step |
|---|---|---|---|---|---|---|
| | | | Best Metric(Best Global Step) | | | |
| **Qwen3-30B-A3B-ReasoningSFT, mini_step=4, max_global_step=180** | | | | | | |
| GSPO | 67.50(35) | 48.33(20) | 94.21(60) | 91.79(35) | 74.62(55) | 90 |
| GRPO+R3 | **69.06**(165) | **51.66**(155) | **95.15**(160) | **93.28**(145) | **77.00**(165) | - |
| **Qwen3-30B-A3B-ReasoningSFT, mini_step=1, max_global_step=180** | | | | | | |
| GRPO | 68.54(25) | 48.95(25) | 93.90(30) | 91.23(20) | 75.30(25) | 40 |
| GRPO+R3 | **69.47**(85) | **53.12**(180) | **95.46**(170) | **93.28**(180) | **76.79**(180) | - |

Table 10: Evaluation results of the RL training process of Qwen3-30B-A3B-ReasoningSFT

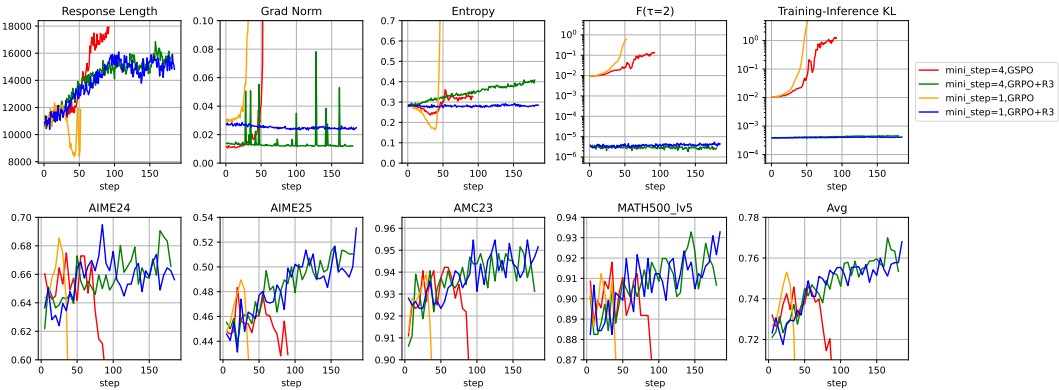

Figure 10: The detailed training metrics and evaluation results of the RL training process of Qwen3-30B-A3B-ReasoningSFT

## J DATASET FILTERING CONFIGURATION

We filtered the math-problem dataset through multiple stages to construct a high-quality subset for programmatic verification and training.

- **Weak-Model Filtering.** We applied `DeepSeek-R1-Distilled-Qwen-7B` (Guo et al., 2025) to perform 16 rounds of filtering. Problems with a pass rate greater than 90% under this weak-model evaluation were removed.

- **Strong-Model Filtering.** We applied the stronger `DeepSeek-R1-0528` (Guo et al., 2025) models for one round of filtering and collected problems that the strong model could not solve.

- **Problem Rewriting.** For problems that failed the strong-model filtering step, we suspect that some failures are due to formatting issues or proof-style requirements. We therefore used several state-of-the-art models (for example, `DeepSeek-V3`)(Liu et al., 2024) to attempt automatic rewriting of these problems into forms that can be programmatically verified.

- **Refiltering.** Problems that were rewritten in the previous step were re-evaluated using both the weak-model and strong-model filtering procedures; problems that then passed the filters were removed from the failure set.

## K    PROBLEM VERIFIER CONFIGURATION

The verifier is designed to extract a final parseable mathematical expression from the model output and compare it to the reference solution, supporting both purely numeric answers and symbolic expressions.

- **Output Format.** The model is prompted to perform step-by-step reasoning and to place the final answer inside a boxed expression. The prompt template (presented here as a Python f-string) is:

  ```
  f"{problem}\nPlease reason step by step,
  and put your final answer within \\boxed{}."
  ```

- **Verification Tool.** We use the open-source `Math-Verify` library[4] to extract the last parseable mathematical expression from the model's answer and compare it against the reference solution. This verifier configuration supports both pure numeric answers and expressions containing variables by comparing semantic mathematical expressions rather than performing raw string matching.

## L    TOKEN TYPE ANALYSIS OF ROUTING DISCREPANCIES

We conducted an additional analysis focused on quantifying the routing discrepancy at the token level.

Specifically, we evaluated the Qwen3-30B-A3B model on a set of 2,048 mathematical problems. During inference, we recorded the routing distribution for each generated token. We then fed the same token sequences through the training framework in a forward pass (without R3), again collecting routing distribution for every token.

For each token, we computed its routing discrepancy, defined as the difference in selected experts between the two passes. The discrepancy ranges from 0 to $L \times K$, where $L$ denotes the number of MoE layers and $K$ represents the number of activated experts. Afterward, we filtered out low-frequency tokens from vocabulary (frequency threshold: 0.02%) and computed the mean discrepancy for the remaining tokens. The top 50 tokens with the highest mean discrepancy are shown in Table 11.

In Figure 11, we visualize the token-level routing discrepancy of a response segment, with red indicating greater routing differences.

A clear pattern emerges: tokens corresponding to mathematical computation exhibit substantially higher routing discrepancy compared to others.

One possible explanation is that performing mathematical computation requires the model to simulate complex reasoning steps in the hidden space. During this process, the hidden states change more sharply, which makes the routing decisions less stable and leads to larger differences between training and inference.

In short, the discrepancy does not happen evenly across all tokens. It mainly appears in certain types of tokens–especially those involving mathematical computation–where the difference becomes noticeably larger.

---

[4]https://github.com/huggingface/Math-Verify

| Rank | Token ID | Token String | Mean Token-Level Diff Router |
|---|---|---|---|
| 1 | 15 | `"0"` | 16.9834 |
| 2 | 37018 | `"frac"` | 16.6235 |
| 3 | 248 | (unvisualizable) | 16.5253 |
| 4 | 50853 | `"cdot"` | 15.9350 |
| 5 | 230 | (unvisualizable) | 15.8782 |
| 6 | 20 | `"5"` | 15.7010 |
| 7 | 116 | (unvisualizable) | 15.6420 |
| 8 | 18 | `"3"` | 15.6307 |
| 9 | 23 | `"8"` | 15.5839 |
| 10 | 19 | `"4"` | 15.3182 |
| 11 | 21 | `"6"` | 15.2691 |
| 12 | 17 | `"2"` | 15.2680 |
| 13 | 22 | `"7"` | 15.0927 |
| 14 | 3245 | `" least"` | 15.0805 |
| 15 | 16 | `"1"` | 14.8400 |
| 16 | 94 | (unvisualizable) | 14.6765 |
| 17 | 258 | `"in"` | 14.4592 |
| 18 | 15896 | (unvisualizable) | 14.3470 |
| 19 | 24 | `"9"` | 14.3465 |
| 20 | 104 | (unvisualizable) | 13.2339 |
| 21 | 79075 | `"boxed"` | 12.9715 |
| 22 | 944 | `"'t"` | 12.6392 |
| 23 | 31807 | `"_2"` | 12.6165 |
| 24 | 26888 | `"sqrt"` | 11.9509 |
| 25 | 15940 | `"sin"` | 11.5561 |
| 26 | 15170 | `"}{"` | 11.4702 |
| 27 | 35702 | `"{\\"` | 9.8007 |
| 28 | 9407 | `"cos"` | 9.6640 |
| 29 | 14 | `"/"` | 8.7482 |
| 30 | 72 | `"i"` | 8.7010 |
| 31 | 62 | `"_"` | 8.5313 |
| 32 | 272 | `" c"` | 8.4916 |
| 33 | 90 | `"{"` | 8.4155 |
| 34 | 89 | `"z"` | 8.1327 |
| 35 | 33 | `"B"` | 8.0483 |
| 36 | 2086 | `" second"` | 7.9709 |
| 37 | 11067 | `" sides"` | 7.8640 |
| 38 | 77 | `"n"` | 7.8423 |
| 39 | 34 | `"C"` | 7.7431 |
| 40 | 88 | `"y"` | 7.6464 |
| 41 | 89669 | `"_0"` | 7.5362 |
| 42 | 48245 | `"π"` | 7.5002 |
| 43 | 12 | `"-"` | 7.4991 |
| 44 | 65 | `"b"` | 7.4563 |
| 45 | 10 | `"+"` | 7.4212 |
| 46 | 1242 | `"sum"` | 7.4100 |
| 47 | 425 | `" B"` | 7.3921 |
| 48 | 74 | `"k"` | 7.3901 |
| 49 | 30986 | `"_1"` | 7.3890 |
| 50 | 198 | `"\n"` | 7.3686 |

Table 11: The top 50 tokens ranked by routing discrepancy

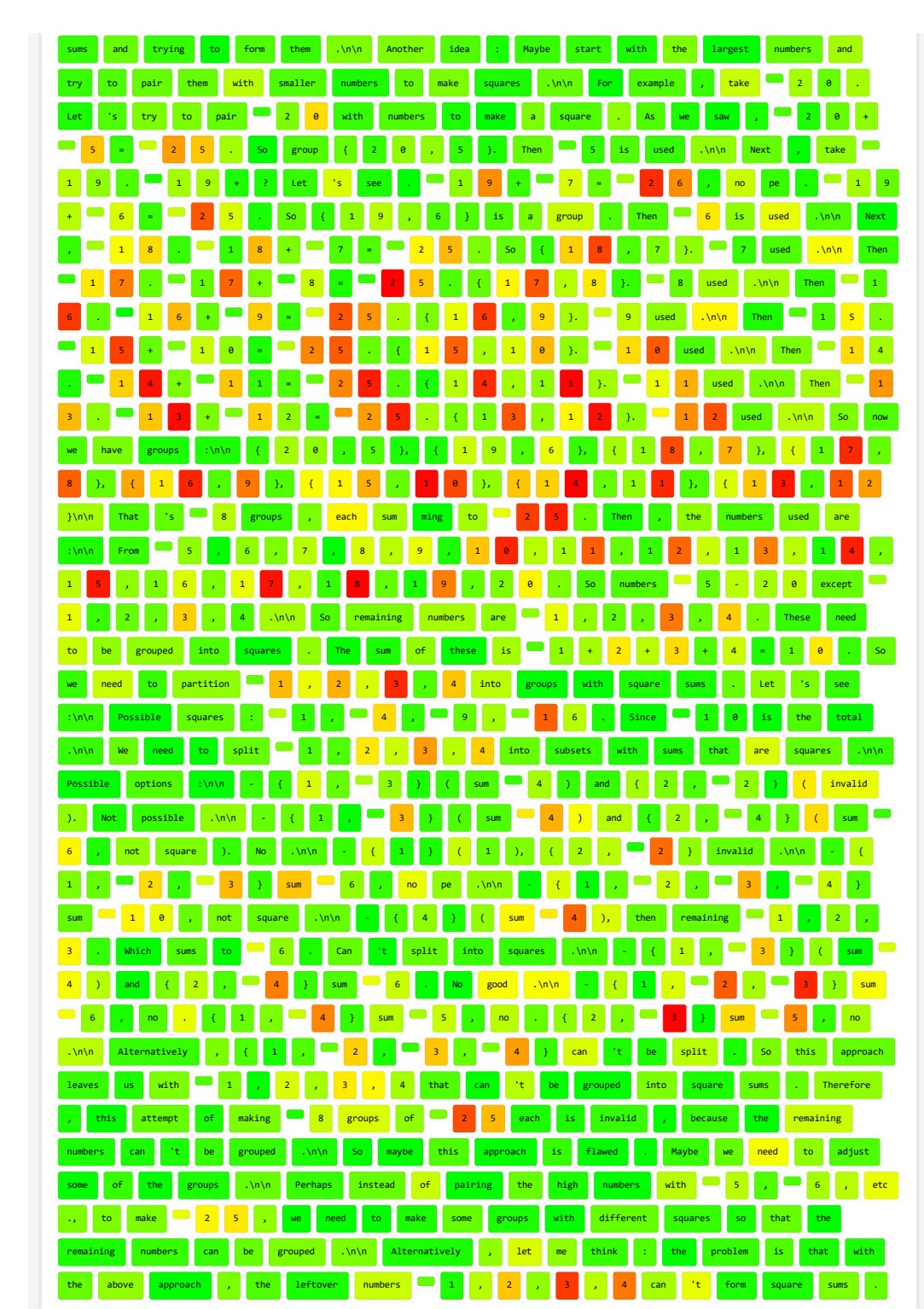

Figure 11: Visualization of routing discrepancy for the response segment, where red indicates large routing discrepancy and green indicates small routing discrepancy

## M   TRAINING HYPERPARAMETERS

| Hyperparameter | Value |
|---|---|
| Batch Size | 256 |
| Group Size | 8 |
| Mini Step | 8 or 1 |
| Max Prompt Length | 2048 |
| Max Response Length | 30720 |
| Optimizer | AdamW |
| Weight Decay | 0.1 |
| Learning Rate | $3 \times 10^{-6}$ or $1 \times 10^{-6}$ |
| Learning Rate Scheduler | Constant |
| Beta1 | 0.9 |
| Beta2 | 0.999 |
| Epsilon | $1 \times 10^{-8}$ |
| Max Gradient Norm | 1.0 |
| DAPO Clip-low | 0.20 |
| DAPO Clip-high | 0.27 |
| GSPO Clip-low | 3e-4 |
| GSPO Clip-high | 4e-4 |
| TIS Clip | 2.0 |

Table 12: Default Training hyperparameters of Reinforcement Learning.

## N   THE USE OF LARGE LANGUAGE MODELS IN THIS PAPER

In this paper, the large language model was only used for polishing the text; it did not contribute to any research ideas or experimental results.

