# OpenReview forum: "Stabilizing MoE Reinforcement Learning by Aligning Training and Inference Routers"
_ICLR.cc/2026/Conference — Submitted to ICLR 2026_

### Official Review · Reviewer_Q997 · 2025-10-30

**Soundness:** 3
**Presentation:** 3
**Contribution:** 2
**Rating:** 6
**Confidence:** 4

**Summary:**

This paper addresses an instability issue in RL for MoE models, that stems from the slight differences that can exist between training and sampling policy. The authors identify that the primary cause is a discrepancy in routing decisions between the inference and training phases, and even inconsistencies within the same training framework across repeated forward passes. To combat this, they propose Rollout Routing Replay (R3), a novel method that captures inference-time routing distributions and replays them during training. R3 aims to align expert selection between phases while preserving gradient flow. Extensive experiments on mathematical reasoning tasks with MoE models demonstrate that R3 substantially reduces training-inference KL divergence, mitigates probability discrepancies, improves training stability and outperforms existing stabilization techniques like GSPO and TIS.

**Strengths:**

- The paper identifies precisely the core problem: routing inconsistencies in MoE models during RL training. It quantifies these discrepancies using KL divergence, an Extreme Token Distribution Function, and a multi-level analysis (router-level, token-level, sequence-level discrepancies, as shown in Figures 2 and 3). This systematic breakdown provides a strong foundation for the proposed solution.
- R3 is a simple yet effective mechanism. By explicitly reusing inference-time routing masks, R3 directly addresses the alignment issue without disrupting gradient flow. The use of router mask caching makes it computationally inexpensive.
- The experimental results are convincing: R3 significantly reduces KL divergence and the frequency of extreme tokens to levels comparable to dense models. Table 1 clearly demonstrates better performance across various benchmarks. The method's applicability to both on-policy and mini-batch style off-policy RL scenarios, as well as its testing across multi-/single-mini-step settings and different model types (SFT/Base), highlights its robustness. R3 is orthogonal to and can be combined with existing optimizers like GRPO and GSPO, and often improves them.
- The most important finding is that R3 contributes to a more stable optimization process. This suggests a healthier and more efficient learning trajectory.

**Weaknesses:**

- Limited explanation of root causes for internal discrepancies: While the paper thoroughly documents the external training-inference discrepancy, and notes that "even multiple runs of the same training framework can produce divergent token probabilities" (L139-140) and "even when the input sequence is identical, the final output probabilities from two forward passes may differ" within Megatron (L224-226), the underlying technical reasons for this internal inconsistency are not deeply explored. A brief discussion on potential causes like floating-point non-determinism, different hardware acceleration paths, or subtle differences in framework execution for "old" vs "new" policy computation would strengthen this point.
- The paper does not provide any explanation on why R3 works better than TIS.
- In general, the related work section would gain from being expanded, and provides only two references related to discrepancies of training frameworks.
- The paper implicitly assumes that inference-time routing decisions are inherently "better" or more stable than what the training framework might produce. A brief justification for this assumption, or a discussion of scenarios where I_infer itself might be problematic (e.g., if the inference engine's router is poorly optimized or prone to its own kind of noise), would be valuable.

**Questions:**

- Would the training still collapse if the MoE router was frozen? This would be an important baseline to add.
- How does the π_train(θ_old) term in the PPO objective (Equation 1) interact with R3's mechanism?
- Beyond the aggregate performance, were there any qualitative observations regarding which types of tokens or which specific layers experienced the most significant reduction in routing discrepancies with R3?
- Can the authors elaborate on why R3 works better than TIS?
- While mask caching is mentioned for efficiency, a more concrete discussion on the memory and computational overhead associated with storing and retrieving these masks for long sequences, especially with many MoE layers and large batch sizes, would be helpful. Is the overhead negligible across all tested scales?
- There is a typo line 155, $\pi_{infer}$ does not appear in the formula
- Could you add a discussion to your paper? How does this paper influence future research?

---

> ### Author Response · Authors · 2025-11-30
> **Response to your questions and weaknesses (part 1)**
>
> Thank you very much for your thoughtful review and the positive comments on our work. We truly appreciate your recognition of our problem formulation, methodology, and experimental design. We are grateful for the time and effort you put into identifying weaknesses and raising clear questions. These comments helped us refine our explanations, expand discussions, and improve clarity throughout the paper. **Your question about the routing discrepancy behavior of different token types was exceptionally insightful, and it led us to investigate further, where we uncovered new and interesting patterns that enriched the contribution of our work.** We have carefully addressed each point in the following response and sincerely appreciate your constructive guidance.
>
>
> # 1. [W1] Limited explanation of root causes for internal discrepancies.
>
> ## Floating-point addition is non-associative
>
> Floating-point addition is non-associative, meaning that the final numerical result may vary depending on the order of operations. The reason is that floating-point numbers have limited precision, and every arithmetic step introduces rounding. For instance, evaluating 1e20 + (-1e20) + 1 yields a different result than 1e20 + 1 + (-1e20). The first one will yield 1 and the second one will yield 0. The reason is that 1e20 + 1 is rounded to 1e20 for the second expression, so the intermediate result loses the 1. This behavior becomes especially noticeable in iterative computations or parallel reductions, where the execution order is not guaranteed.
>
> ## Floating-Point Atomic Operations
>
> When atomic addition is applied to floating-point values, the execution order of concurrent updates cannot be strictly guaranteed, leading to non-deterministic and unreproducible outcomes. A representative case is the scatter_add_ operator in PyTorch, which allows repeated indices in the "index" tensor. When multiple values are accumulated into the same memory address, the summation order depends on the hardware scheduling behavior rather than a fixed rule, and therefore the output cannot be guaranteed to be identical across runs.
>
> ## No Batch Invariance
>
> Another source of discrepancy arises when the model processing pipeline does not satisfy batch invariance. In inference framework, new sequences continuously enter the batch while completed sequences exit. Because the batch composition changes dynamically, an identical input sequence may occupy different positions in the batch across multiple forward passes. The blog [1] has pointed out that this dynamic reordering may lead to discrepancies in output probability distributions.
>
> This phenomenon may also affect the training framework. For instance, when computing the outputs of an old policy and a new policy on the same dataset--common in reinforcement learning or PPO-style optimization--differences in sample order within a batch may yield statistically significant deviations, even though the model parameters are identical.
>
> ## Variation in Operator and Kernel Selection
>
> Variation in operator and kernel selection can lead to numerical differences between training and inference framework. Differences in operator implementations across frameworks may change the reduction order in floating-point operations, resulting in small deviations. Additionally, even when high-level frameworks such as PyTorch appear to use the same operators, different batch sizes may trigger different backend kernels. For example, matrix multiplication operations on gpu may select various backend kernels depending on matrix size, data type, batch configuration, and hardware architecture to optimize performance. While these kernels are mathematically equivalent, dynamic scheduling can modify the sequence of floating-point accumulation. Consequently, different runtime conditions or batch settings may introduce slight numerical divergence.
>
> ---
>
> Reference:
>
> [1]. He, Horace and Thinking Machines Lab, "Defeating Nondeterminism in LLM Inference", Thinking Machines Lab: Connectionism, Sep 2025. URL: https://thinkingmachines.ai/blog/defeating-nondeterminism-in-llm-inference/
>
> ---
>
> We will include these analyses in the revised version of our paper.

---

> ### Author Response · Authors · 2025-11-30
> **Response to your questions and weaknesses (part 2)**
>
> # 2. [W2 & Q4] Can the authors elaborate on why R3 works better than TIS?
>
> **TIS just suppresses the symptoms, whereas R3 corrects the root cause**
>
> Thank you for this insightful question. The key difference is that TIS only adjusts the loss by applying the importance sampling ratio, but it does not correct the forward routing distribution that generated such instabilities. Thus, gradients under TIS are still distributed to experts that were not used during inference, which means the gap between training and inference remains.
>
> In contrast, R3 modifies both the forward pass and the backward routing by replaying the exact routing distribution from inference. This ensures that gradients flow exclusively to the experts that were actually responsible for generating the rollout tokens. As a result, R3 directly eliminates the root cause of mismatch rather than mitigating it by importance sampling ratio. Empirically, this leads to stronger improvements in stability and performance.
>
> In short: TIS just suppresses the symptoms, whereas R3 corrects the root cause.
>
>
> # 3. [W3] The related work section would gain from being expanded.
>
> Thank you for your suggestion! We will expand work related to training–inference discrepancies, nondeterminism, and floating-point precision. The following works will be included into the related work section (some of which have already been cited in the paper):
>
> [1]. MiniMax, et al. (2025). MiniMax-M1: Scaling Test-Time Compute Efficiently with Lightning Attention. arXiv. https://arxiv.org/abs/2506.13585
>
> [2]. He, H., et al. (2025). Defeating Nondeterminism in LLM Inference. Thinking Machines Lab (Blog). https://thinkingmachines.ai/blog/defeating-nondeterminism-in-llm-inference
>
> [3]. Yao, F., et al. (2025). Your Efficient RL Framework Secretly Brings You Off-Policy RL Training. Notion. https://fengyao.notion.site/off-policy-rl
>
> [4]. Liu, J., et al. (2025). When Speed Kills Stability: Demystifying RL Collapse from the Training-Inference Mismatch. https://yingru.notion.site/When-Speed-Kills-Stability-Demystifying-RL-Collapse-from-the-Training-Inference-Mismatch-271211a558b7808d8b12d403fd15edda
>
> [5]. Zhao, X., et al. (2025). Small Leak Can Sink a Great Ship — Boost RL Training on MoE with IcePop! Notion. https://ringtech.notion.site/icepop
>
> [6]. Qi, P., et al. (2025). Defeating the Training-Inference Mismatch via FP16. arXiv. https://arxiv.org/abs/2510.26788
>
> [7]. Atil, B., et al. (2024). Non-Determinism of "Deterministic" LLM Settings. arXiv. https://arxiv.org/abs/2408.04667
>
> [8]. Yuan, J., et al. (2025). Understanding and Mitigating Numerical Sources of Nondeterminism in LLM Inference. OpenReview / NeurIPS. https://openreview.net/forum?id=Q3qAsZAEZw
>
> # 4. [W4] Implicitly assumes that inference-time routing decisions are inherently "better" or more stable.
>
> Thank you for your suggestion.
>
> What we want to clarify is that inference-time routing appears “better” because in reinforcement learning the training sequences are generated by the inference framework itself.
>
> For supervised learning, where data come from an external distribution, the routing distribution during training framework and inference framework (although SFT doesn't need inference framework) can be considered symmetric, and neither is inherently preferable.
>
> However, in reinforcement learning, the generated sequences are produced by the model’s own inference process. This means the inference framework determines which experts actually contributed to producing the tokens, while the training stage updates parameters based on these same sequences.
>
> Under this setting, the inference-time routing is not just one of two equivalent routing choices, it represents the actual decision path that produced the training sequences. Since every token in the rollout sequences is generated under that routing pattern, reusing the inference routing during training ensures that gradients update only the experts that were truly involved in sequence generation. This makes the optimization process better aligned with the model’s real behavior. In contrast, if training continues to use a routing pattern unrelated to inference, a train–infer mismatch emerges, leading to unstable updates, inflated importance sampling weights, and possible policy collapse.
>
> ---
>
> We will include these contents and analyses in the revised version of our paper.

---

> ### Author Response · Authors · 2025-11-30
> **Response to your questions and weaknesses (part 3)**
>
> # 5. [Q1] Would the training still collapse if the MoE router was frozen?
>
> **Yes**
>
> Thank you for your question. We conducted experiments on the Qwen3-30B-A3B-Base model. In these experiments, we froze the model’s router and trained it using the same settings as described in the paper. The results show that the model still **collapsed after 120 training steps**, and its performance improvement was significantly slower compared to the version without router freezing.
>
> | Training Steps | Avg Val Score |
> |----------------|---------------|
> | 0              | 39.31         |
> | 10             | 41.05         |
> | 20             | 42.91         |
> | 30             | 43.97         |
> | 40             | 44.73         |
> | 50             | 46.84         |
> | 60             | 48.72         |
> | 70             | 54.65         |
> | 80             | 58.16         |
> | 90             | 60.49         |
> | 100            | 62.91         |
> | 110            | 54.02         |
> | 120            | 25.12 (crash) |
>
> These findings indicate that the collapse of the model cannot be attributed solely to the failure of the router parameters. Instead, it is highly likely that the collapse involves the degradation of the model parameters as a whole.
>
> # 6. [Q2] How does the π_train(θ_old) term in the PPO objective (Equation 1) interact with R3's mechanism?
>
> Thank you for your question. In most open-source large-model reinforcement learning frameworks(e.g. VeRL), the PPO algorithm involves three forward passes: (1) an **inference stage** (computing $\pi_{\text{infer}}(\theta_{old})$ using the inference framework), (2) a **recomputation stage** (computing $\pi_{\text{train}}(\theta_{old})$ using the training framework), and (3) an **update stage** (computing $\pi_{\text{train}}(\theta)$ using the training framework).
>
> The R3 method reuses the routing computed during the inference stage for both the recomputing stage and update stage, as shown in the left panel of Figure 1. This means that both $\pi_{\text{train}}(\theta_{old})$ and $\pi_{\text{train}}(\theta)$ are computed using the routing generated during the inference stage.
>
> When computing the loss we still use the standard PPO objective, i.e.:
>
> $$
> L_{\text{PPO}} = \frac{1}{T}\sum_{t=1}^{T}
> \min\left(
> \frac{\pi_{\text{train}}(\theta)(t)}{\pi_{\text{train}}(\theta_{old})(t)} \cdot A_t,\
> \text{clip}\left(
> \frac{\pi_{\text{train}}(\theta)(t)}{\pi_{\text{train}}(\theta_{old})(t)},\
> 1-\epsilon,\
> 1+\epsilon
> \right)\cdot A_t
> \right)
> $$

---

> ### Author Response · Authors · 2025-11-30
> **Response to your questions and weaknesses (part 4)**
>
> # 7. [Q3] Were there any qualitative observations regarding which types of tokens or which specific layers experienced the most significant reduction in routing discrepancies with R3?
>
> Thank you for raising this point. To address it, we conducted an additional analysis focused on quantifying the routing discrepancy at the token level.
>
> Specifically, we evaluated the Qwen3-30B-A3B model on a set of 2,048 mathematical problems. During inference, we recorded the routing distribution for each generated token. We then fed the same token sequences through the training framework in a forward pass (without R3), again collecting routing distribution for every token.
>
> For each token, we computed its routing discrepancy, defined as the difference in selected experts between the two passes. The discrepancy ranges from 0 to $L \times K$, where $L$ denotes the number of MoE layers and $K$ represents the number of activated experts. Afterward, we filtered out low-frequency tokens from vocabulary (frequency threshold: 0.0002) and computed the mean discrepancy for the remaining tokens. The top 50 tokens with the highest mean discrepancy are shown below.
>
> | Rank | Token ID | Token String | Mean Token-Level Diff Router |
> |-|-|-|-|
> | 1 | 15 | "0" | 16.98344897470902 |
> | 2 | 37018 | "frac" | 16.623588282308788 |
> | 3 | 248 | "\ufffd" | 16.52533183772668 |
> | 4 | 50853 | "cdot" | 15.935015506061461 |
> | 5 | 230 | "\ufffd" | 15.878215223097113 |
> | 6 | 20 | "5" | 15.701047474965781 |
> | 7 | 116 | "\ufffd" | 15.642046015601856 |
> | 8 | 18 | "3" | 15.630736091752944 |
> | 9 | 23 | "8" | 15.583904557058718 |
> | 10 | 19 | "4" | 15.318261976134123 |
> | 11 | 21 | "6" | 15.269115372776193 |
> | 12 | 17 | "2" | 15.268020624730564 |
> | 13 | 22 | "7" | 15.092727710490228 |
> | 14 | 3245 | " least" | 15.080556193978396 |
> | 15 | 16 | "1" | 14.84008687920255 |
> | 16 | 94 | "\ufffd" | 14.676510780443365 |
> | 17 | 258 | "in" | 14.459214501510575 |
> | 18 | 15896 | "\ufffd" | 14.347073942269672 |
> | 19 | 24 | "9" | 14.346531580686175 |
> | 20 | 104 | "\ufffd" | 13.233911736576035 |
> | 21 | 79075 | "boxed" | 12.971563303801085 |
> | 22 | 944 | "'t" | 12.639209738264691 |
> | 23 | 31807 | "₂" | 12.616588890561493 |
> | 24 | 26888 | "sqrt" | 11.950984236530468 |
> | 25 | 15940 | "sin" | 11.556118827942816 |
> | 26 | 15170 | "}{" | 11.470249961662322 |
> | 27 | 35702 | "{\\" | 9.800712347354139 |
> | 28 | 9407 | "cos" | 9.664099272571674 |
> | 29 | 14 | "/" | 8.748205530719266 |
> | 30 | 72 | "i" | 8.70107948969578 |
> | 31 | 62 | "_" | 8.531319310295485 |
> | 32 | 272 | " c" | 8.491600559962668 |
> | 33 | 90 | "{" | 8.415587685225148 |
> | 34 | 89 | "z" | 8.132773508674642 |
> | 35 | 33 | "B" | 8.048343344313118 |
> | 36 | 2086 | " second" | 7.9709906542056075 |
> | 37 | 11067 | " sides" | 7.864038545773106 |
> | 38 | 77 | "n" | 7.84231792964115 |
> | 39 | 34 | "C" | 7.743140780114125 |
> | 40 | 88 | "y" | 7.646424112716248 |
> | 41 | 89669 | "₀" | 7.536286664650741 |
> | 42 | 48245 | "π" | 7.500212554910019 |
> | 43 | 12 | "-" | 7.499194847020934 |
> | 44 | 65 | "b" | 7.456314606741573 |
> | 45 | 10 | "+" | 7.421206695654645 |
> | 46 | 1242 | "sum" | 7.4100188711614345 |
> | 47 | 425 | " B" | 7.3921987798260185 |
> | 48 | 74 | "k" | 7.390182903446321 |
> | 49 | 30986 | "₁" | 7.389045682850993 |
> | 50 | 198 | "\n" | 7.368648786230155 |
>
> A clear pattern emerges: tokens corresponding to mathematical computation exhibit substantially higher routing discrepancy compared to others.
>
> One possible explanation is that performing mathematical computation requires the model to simulate complex reasoning steps in the hidden space. During this process, the hidden states change more sharply, which makes the routing decisions less stable and leads to larger differences between training and inference.
>
> In short, the discrepancy does not happen evenly across all tokens. It mainly appears in certain types of tokens--especially those involving mathematical computation--where the difference becomes noticeably larger.

---

> ### Author Response · Authors · 2025-11-30
> **Response to your questions and weaknesses (part 5)**
>
> # 8. [Q5] Discussion on the memory and computational overhead associated with storing and retrieving these masks?
>
> ## Routing-mask caches in Inference Framework Comparing with KVCache
>
> **For inference framework, Router mask cache is stored in CPU memory and is only 1/128 the size of the KV Cache on the GPU.**
>
> Thank you for your question. We store routing masks along with KVCache in the inference engine, with KVCache on GPU HBM and routing masks on CPU memory. As shown in the table below, the memory usage of routing mask per token is far less than KVCache. The datatype of KVCache is bfloat16 (2 bytes), and the datatype of routing mask is int16 (2 bytes). Assume we have 40GB each GPU for KVCache prefix caching, we only need 2.5GB memory per node to store the routing masks of the tokens on 8 GPUs, which is negligible in deployment.
>
> | \ | KVCache | Routing Mask Cache |
> |---|---|---|
> | Each Token | DataType * #Layer * (KDim+VDim) | DataType * #Layer * Topk |
> |Each Token in Qwen3-30B-A3B | 2 * 48 * (512 + 512) | 2 * 48 * 8 |
> |437K tokens per GPU | 40GB on GPU HBM | 0.3GB on CPU memory |
> |437K*8 tokens per node (8 GPUs) | 40GB * 8 on HBM of 8 GPUs | 2.5GB on CPU memory |
>
> ## Routing Mask in Training Framework per Global Training Step
>
> **For our configuration (B = 256), (G = 8), (N = 32768), (L = 48), and (K = 8), the max memory of routing mask is approximately 48 GB in Training Framework**.
>
> Here, we also provide a detailed calculation of the routing mask memory footprint per training step.
>
> Assume the number of prompts is (B), the number of groups per prompt is (G), the max sequence length is (N), the MoE model contains (L) layers, and each layer selects (K) experts. Since we store the selected expert indices in int16, the max routing mask size is:
>
> $$
>  \text{Mask Size} = 2 \times B \times G \times N \times L \times K \text{ bytes}
> $$
>
> For the configuration (B = 256), (G = 8), (N = 32768), (L = 48), and (K = 8), the max memory requirement is approximately 48 GB.
>
> In practice, these masks are distributed across the training infrastructure rather than being stored on a single machine. Moreover, they are typically in CPU memory and only loaded into GPU memory on demand during training.
>
> If the router states are distributed across T training nodes--for example, **if T = 32--then each machine stores only around 1.5 GB** of mask data. For modern AI clusters equipped with high-bandwidth interconnects and large memory capacity, both the storage and communication overhead introduced by these routing masks are well within operational limits.
>
> We hope this clarification demonstrates that routing mask caching is practical and scalable under realistic deployment settings.
>
> # 9. [Q6] There is a typo line 155, I_infer does not appear in the formula
>
> We appreciate the reviewers for carefully examining the manuscript. We apologize for the oversight and confirm that Equation (2) contains a typographical error. Specifically, the denominator should be $\pi_{\text{infer}}$ instead of $\pi_{\text{train}}$. We have corrected this issue, and the updated equation (2) is now:
>
> $$
> D_{\text{KL}}(\pi_{\text{train}}(\theta)|| \pi_{\text{infer}}(\theta))
> \approx \frac{1}{|T|}\sum_{t \in T}\bigl[\frac{\pi_{\text{train}}(\theta)(t)}{\pi_{\text{infer}}(\theta)(t)}-1 - \log \frac{\pi_{\text{train}}(\theta)(t)}{\pi_{\text{infer}}(\theta)(t)}\bigr].
> $$

---

### Official Review · Reviewer_SRHz · 2025-10-31

**Soundness:** 3
**Presentation:** 3
**Contribution:** 3
**Rating:** 4
**Confidence:** 3

**Summary:**

This paper addresses a key source of instability in reinforcement learning (RL) for Mixture-of-Experts (MoE) large language models: routing inconsistency between inference (rollout) and training. The authors observe that even under identical conditions, MoE routers can select different experts across passes, creating divergence between rollout and training policies that can trigger catastrophic collapse. To mitigate this, the paper proposes Rollout Routing Replay (R3),  a simple yet effective mechanism that records routing distributions (expert selections) from the inference engine and reuses them in the training phase.

This alignment substantially reduces training–inference KL divergence, lowers the number of “extreme tokens", and improves training stability without slowing down throughput. Empirical results on large MoE models (e.g., Qwen3-30B-A3B) show that R3 outperforms GSPO and TIS in terms of stability and final performance on math RL tasks.

**Strengths:**

- The paper isolates a concrete but under-explored source of RL collapse: router nondeterminism in MoE models.
- The proposed fix, replaying inference routing distributions into training, is conceptually simple yet novel and directly addresses the identified failure mode.
- The empirical evaluation demonstrates evidence of reduced policy KL and improved stability.
- The method integrates seamlessly with existing RL frameworks (e.g., SGLang rollout + Megatron training) and appears computationally lightweight.
- Comparisons against strong baselines (GSPO, TIS) show consistent improvements.
- The paper provides intuitive metrics (KL divergence, extreme-token statistics) to support its argument.
- Stabilizing MoE RL is a highly relevant and urgent problem in LLM post-training.

**Weaknesses:**

- The PPO and KL divergence equations contain inconsistent notation (π_train appears twice where π_infer should).
- Equation (2) appears malformed and should be corrected and verified.
- All experiments are on math RL tasks. Additional evidence on other domains (e.g., code RLVR, reasoning, dialogue) would increase generality.
- The figures seem to use best checkpoints from single runs. Multi-seed mean ± CI and last-step metrics are required for reliability.
- The paper claims “no slowdown,” but lacks any wall-time, throughput, or memory-usage comparison. Quantitative data is needed.
- No analysis of (i) old-policy vs. update-policy replay, (ii) mask staleness, or (iii) top-K sensitivity.
- The dataset filtering (100 k math problems) and verifier configuration are insufficiently documented.
- Hyperparameter tables, seeds, and scripts should be released for full reproducibility.

**Questions:**

- Please confirm and fix Equation (2). Is the KL computed per-token or per-sequence?
- Quantify GPU-hour and tokens/s differences with and without R3. How large are the routing-mask caches?
- In multi-mini-step updates, how long can stored routing masks remain valid before benefits degrade?
- How does R3 interact with load-balancing or entropy penalties commonly applied to routers?
- Does R3 help (or hurt) in non-math RL tasks or dense models with trivial routing?
- Are improvements over GSPO/TIS statistically significant across seeds? Please report mean ± std.
- Include a short analysis showing how R3 reduces gradient variance or stabilizes importance ratios.

**Details Of Ethics Concerns:**

There are no ethics concerns to report.

---

> ### Author Response · Authors · 2025-11-27
> **Response to your questions and weaknesses (part 1)**
>
> We sincerely appreciate the reviewers for taking the time to evaluate our submission and provide thoughtful comments. Your feedback will significantly help us improve the quality and clarity of our manuscript. Below, we provide detailed responses to each of your weaknesses and questions.
>
> # 1. [Q1 & W1 & W2] Please confirm and fix Equation (2). Is the KL computed per-token or per-sequence?
> We appreciate the reviewers for carefully examining the manuscript. We apologize for the oversight and confirm that Equation (2) contains a typographical error. Specifically, the denominator should be $\pi_{\text{infer}}$ instead of $\pi_{\text{train}}$. We have corrected this issue, and the updated equation (2) is now:
>
> $$
> D_{\text{KL}}(\pi_{\text{train}}(\theta)|| \pi_{\text{infer}}(\theta))
> \approx \frac{1}{|T|}\sum_{t \in T}\bigl[\frac{\pi_{\text{train}}(\theta)(t)}{\pi_{\text{infer}}(\theta)(t)}-1 - \log \frac{\pi_{\text{train}}(\theta)(t)}{\pi_{\text{infer}}(\theta)(t)}\bigr].
> $$
>
> We would like to clarify that the KL divergence is computed as a **token-level** average, rather than a sequence-level average.

---

> ### Author Response · Authors · 2025-11-27
> **Response to your questions and weaknesses (part 2)**
>
> # 2. [Q2 & W5] Quantify GPU-hour and tokens/s differences with and without R3. How large are the routing-mask caches?
>
> ## 2.1 Quantify GPU-hour and tokens/s differences with and without R3
>
> **R3 method only introduced a 3.45% overhead in runtime.**
>
> Thank you for your suggestion. We conduct end-to-end performance benchmarking to provide a clear comparison. Specifically, we selected the Qwen-30B-A3B model and performed reinforcement learning training under two settings: with R3 and without R3. The experimental configuration is as follows: batch_size = 256, N=8, max_length = 32K.
>
> To ensure that the measurement results are stable and representative, we adopted the following precautions:
> - The learning rate was set to 0.0 to disable model parameter updates while still executing full backward and optimizer operations;
> - Dynamic sampling was disabled to avoid additional variance during measurement;
> - Evaluation was disabled to remove extra overhead;
> - Training was executed for 15 steps, and only the runtime of steps 5–14 (10 steps in total) was used for statistical averaging.
>
> The measured runtime represents true end-to-end execution latency, including inference, forward passes, backward passes, model optimization, and communication overhead between inference and training framework. The final benchmarking results are summarized in the table below.
>
> | Method | # Token | Runtime(second) | Runtime(hour) | # Token/s | Relative time cost (compared with the baseline) |
> |--------|---------|---------|---------|---------|-----------------------|
> | w/o R3 | 203989395 | 4820 | 1.339 | 42321 | 100% |
> | w/ R3 | 203201992 | 4967 | 1.380 | 40910 | 103.45% |
>
> Our findings indicate that, after careful optimization, the inclusion of R3 results in only a 3.45% reduction in end-to-end training throughput. This overhead is minimal and suggests that R3 can be integrated into real-world systems with negligible performance impact.
> We hope this quantitative analysis addresses the reviewer’s request and demonstrates the practicality of deploying R3 in production environments.
>
> ##  2.2 How large are the routing-mask caches?
>
> ### Routing-mask caches in Inference Framework Comparing with KVCache
>
> **For inference framework, Router mask cache is stored in CPU memory and is only 1/128 the size of the KV Cache on the GPU.**
>
> Thank you for your question. We store routing masks along with KVCache in the inference engine, with KVCache on GPU HBM and routing masks on CPU memory. As shown in the table below, the memory usage of routing mask per token is far less than KVCache. The datatype of KVCache is bfloat16 (2 bytes), and the datatype of routing mask is int16 (2 bytes). Assume we have 40GB each GPU for KVCache prefix caching, we only need 2.5GB memory per node to store the routing masks of the tokens on 8 GPUs, which is negligible in deployment.
>
> | \ | KVCache | Routing Mask Cache |
> |---|---|---|
> | Each Token | DataType * #Layer * (KDim+VDim) | DataType * #Layer * Topk |
> |Each Token in Qwen3-30B-A3B | 2 * 48 * (512 + 512) | 2 * 48 * 8 |
> |437K tokens per GPU | 40GB on GPU HBM | 0.3GB on CPU memory |
> |437K*8 tokens per node (8 GPUs) | 40GB * 8 on HBM of 8 GPUs | 2.5GB on CPU memory |
>
> ### Routing Mask in Training Framework per Global Training Step
>
> **For our configuration B = 256, G = 8, N = 32768, L = 48, and K = 8, the max memory of routing mask is approximately 48 GB in Training Framework**.
>
> Here, we also provide a detailed calculation of the routing mask memory footprint per training step.
>
> Assume the number of prompts is (B), the number of groups per prompt is (G), the max sequence length is (N), the MoE model contains (L) layers, and each layer selects (K) experts. Since we store the selected expert indices in int16, the max routing mask size is:
>
> $$
>  \text{Mask Size} = 2 \times B \times G \times N \times L \times K \text{ bytes}
> $$
>
> For the configuration B = 256, G = 8, N = 32768, L = 48, and K = 8, the max memory requirement is approximately 48 GB.
>
> In practice, these masks are distributed across the training infrastructure rather than being stored on a single machine. Moreover, they are typically in CPU memory and only loaded into GPU memory on demand during training.
>
> If the router states are distributed across T training nodes--for example, **if T = 32--then each machine stores only around 1.5 GB** of mask data. For modern AI clusters equipped with high-bandwidth interconnects and large memory capacity, both the storage and communication overhead introduced by these routing masks are well within operational limits.
>
> We hope this clarification demonstrates that routing mask caching is practical and scalable under realistic deployment settings.

---

> ### Author Response · Authors · 2025-11-27
> **Response to your questions and weaknesses (part 3)**
>
> # 3. [Q3 & W6.2] In multi-mini-step updates, how long can stored routing masks remain valid before benefits degrade?
>
> **Routing mask cache is only valid within each rollout step(global step).**
>
> Thank you for your question. Like KVCache, routing mask cache is only valid within each rollout step(global step). Therefore, in on-policy training, routing mask cache is valid for 1 training mini-step; while in 8-mini-step off-policy training, routing mask cache is valid for 8 training mini-step. In the next rollout step, the routing mask cache and KVCache will be flushed together. Whether using cache only affects efficiency rather than model accuracy, since the same routing mask will simplify be re-computed if no cache available.
>
> # 4. [Q4] How does R3 interact with load-balancing or entropy penalties commonly applied to routers?
>
> Thank you for your question. In this work, we did not apply load balancing in our experiments due to concerns about introducing extra variables and affecting the model's performance.
>
> However, R3 can certainly work with load balancing or entropy penalties if they are used. For example, using an auxiliary loss, its calculation is:
>
> $$
> L_{\text{aux}} = F \cdot P = \sum_{i=1}^{n} F_i P_i
> $$
>
> where P is the probability distribution and F is the chosen router distribution.
>
> Normally, F is the top-k of P under the training framework. When using R3, we can directly set F as the top-k of P in the inference framework. This approach not only aligns with R3's design philosophy, but also may be more reasonable because the goal of load balancing for large models is mainly to balance the computation during inference. Using the inference top-k directly fits this goal better.
>
> # 5. [Q5 & W3] Does R3 help (or hurt) in non-math RL tasks or dense models with trivial routing?
>
> ## 5.1 Does R3 help (or hurt) in non-math RL tasks?
>
> **R3 shows superior performance on the SWE task.**
>
> Thank you for your question. To further strengthen our empirical evidence, we extended our experiments to the Software Engineering (SWE) task. This task has recently become a key benchmark for state-of-the-art models, as it simultaneously evaluates a model’s code generation ability, tool-use capability, and multi-turn reasoning performance. We conducted reinforcement learning experiments under this task to validate both the effectiveness of the proposed R3 method and the feasibility of the Router Mask Cache mechanism described in Section 4.2.
>
> We conduct reinforcement learning experiments using the Qwen3-30B-A3B model. We use R2E-Gym-Lite[1] as the training dataset and SWE-bench Verified[2] as the validation dataset. For training, We set the maximum sequence length to 65536 tokens (including the prompt, model response, and environment observation), the batch size to 64, the maximum number of interaction steps to 50, the learning rate to 2e−6, and the mini_step to 1. The experiments were trained for 180 steps. Evaluation was conducted on the SWE-Bench-Verified benchmark. The experimental results are summarized in the table below and demonstrate that Replay (R3) successfully stabilizes reinforcement learning training on SWE task.
>
> | Method    | SWE-bench-Verified (↑)    | Best Global Step | Crash Step |
> |-----------|---------------------------|------------------|------------|
> | w/o R3    | 31.80                     | 70               | 90         |
> | w/ R3     | 38.60                     | 160              | -          |
>
> We will include these results in the revised version of our paper. We believe these findings provide additional evidence that R3 is not only theoretically motivated but also practical and effective in real-world, high-demand reasoning and software engineering scenarios.
>
>
>
> ## 5.2 Does R3 help (or hurt) in dense models with trivial routing?
>
> **R3 cannot be applied to Dense Models**
>
> R3 cannot be applied to Dense Models because there is no router
> We would like to offer one important clarification: the proposed R3 method is specifically designed for MoE architectures and cannot be applied to dense models. The method relies on properties unique to routed computation and expert selection, which do not exist in dense architectures. Therefore, applying R3 to dense models is outside its intended design scope.
>
> ---
>
> Reference:
>
> Jain, N., et al. (2025). R2e-gym: Procedural environments and hybrid verifiers for scaling open-weights swe agents. arXiv. https://arxiv.org/abs/2504.07164
>
> Jimenez, C. E., et al. (2024). Swe-bench: Can language models resolve real-world GitHub issues? arXiv preprint arXiv. https://arxiv.org/abs/2310.06770

---

> ### Author Response · Authors · 2025-11-27
> **Response to your questions and weaknesses (part 4)**
>
> # 6. [Q6] Are improvements over GSPO/TIS statistically significant across seeds? Please report mean ± std.
>
> Thank you for your question. **Yes! The improvements is statistically significant.**
> We have conducted three runs under different random seeds for GSPO, TIS on Qwen3-30B-A3B model with mini-step=4, learning rate=2e-6. The results are as follows:
>
> | Method | seed | AIME24      | AIME25      | AMC         | Math500(lv5)| Val Score   |
> |--------|------|-------------|-------------|-------------|-------------|-------------|
> | GSPO   | 2025 | 56.35       | 36.56       | 90.47       | 87.13       | 67.63       |
> | GSPO   | 2026 | 45.21       | 35.52       | 85.00       | 80.04       | 61.44       |
> | GSPO   | 2027 | 55.73       | 34.38       | 90.31       | 87.50       | 66.98       |
> | TIS    | 2025 | 47.81       | 33.13       | 86.41       | 84.89       | 63.06       |
> | TIS    | 2026 | 49.58       | 32.50       | 86.56       | 84.33       | 63.24       |
> | TIS    | 2027 | 51.56       | 35.00       | 88.28       | 85.82       | 65.16       |
> | R3     | 2025 | 57.54       | 39.06       | 91.41       | 88.81       | 69.20       |
> | R3     | 2026 | 56.04       | 35.31       | 89.06       | 88.43       | 67.21       |
> | R3     | 2027 | 52.81       | 37.92       | 88.13       | 84.14       | 65.75       |
>
> | Method | Val Score Mean±Std    |
> |--------|-------------|
> | GSPO   | 65.35±3.40  |
> | TIS    | 63.82±1.16  |
> | R3     | 67.39±1.73  |
>
> From the results, we can see that R3 performs than both GSPO and TIS across different random seeds. TIS is quite stable, but its overall performance is always lower than R3. GSPO sometimes fails on certain runs, which lowers its average score and increases its variance, showing that it is not very stable. R3, however, has higher average scores and smaller variance, which means it is more stable and gives more reliable results.
>
> We will include these analyses and results in the revised version of our paper.
>
> # 7. [Q7] Include a short analysis showing how R3 reduces gradient variance or stabilizes importance ratios.
>
> Thank you for your suggestion.
>
> **Stabilizing importance ratios**:
>
> We measured the mean and variance of importance sampling ratios during the first reinforcement learning step, with and without R3, on Qwen3-30B-A3B. The results are shown below:
>
> | Method   | Mean±Std           |
> |--------  |-------------       |
> | w/o R3   | 0.999994±0.056408  |
> | w/ R3    | 0.999995±0.039037  |
>
> We observe that using R3 significantly reduces the variance of the importance sampling ratio. This happens because R3 improves the alignment between training and inference behaviors, which reduces fluctuations in the importance ratios.
>
> **Reducing gradient variance**:
>
> We also observed that, on the Qwen3-30B-A3B-Base model, training without R3 leads to consistently larger gradients. We believe this is because the model produces unstable rollout trajectories, which leads to unstable gradient updates. In contrast, when training with R3, the model quickly adapts to the experts selected during inference. This allows it to identify the correct optimization direction early, generate stable rollout trajectories within only a few steps, and therefore reduce gradient magnitude and stabilize gradient updates much earlier in training.
>
> We will include these analyses in the revised version of our paper.

---

> ### Author Response · Authors · 2025-11-27
> **Response to your questions and weaknesses (part 5)**
>
> # 8. [W6] No analysis of (i) old-policy vs. update-policy replay, (ii) mask staleness, or (iii) top-K sensitivity.
>
> ## 8.1 old-policy vs. update-policy replay
>
> Thank you for your suggestion. **Both old-policy replay and update-policy replay should be applied**
>
> Old-policy (or recompute-old-policy) and update-policy are two different stages in LLM RL training, and both occur within the training framework. In R3, we apply the method in both the recompute-old-policy stage and the update-policy stage. If we apply R3 only in the update-policy stage, the old-policy will no longer be trustworthy, and importance sampling will become unreliable. On the other hand, if we apply R3 only in the old-policy stage, it will not be able to correct the forward and backward computations during model updates, which goes against the original design purpose of R3.
>
> ## 8.2 mask staleness
>
> **Routing mask cache is only valid within each rollout step(global step).**
>
> Thank you for your suggestion. Like KVCache, routing mask cache is only valid within each rollout step(global step). Therefore, in on-policy training, routing mask cache is valid for 1 training mini-step; while in 8-mini-step off-policy training, routing mask cache is valid for 8 training mini-step. In the next rollout step, the routing mask cache and KVCache will be flushed together. Whether using cache only affects efficiency rather than model accuracy, since the same routing mask will simplify be re-computed if no cache available.
>
> ## 8.3 top-K sensitivity
>
> Thank you for your suggestion. We are sorry and we didn't get what your "top-K sensitivity" means.
>
> If "top-K sensitivity" means how the choice of K affects the performance of R3, the number of experts selected by the router is a fixed hyperparameter from the start of pre-training. It cannot be changed later.
>
> If "top-K sensitivity" is about the difference of top-k routing between training and inference frameworks, we have already provided a detailed discussion in Section 3.2, covering the differences at three levels: the router level, the token level, and the sequence level.
>
> If the term "top-K sensitivity" has other meaning, please let us know. We would be happy to clarify and address your question.
>
> # 9. [W7] The dataset filtering (100 k math problems) and verifier configuration are insufficiently documented.
>
> Thank you for your suggestion. We will the details about the dataset filtering and verifier configuration in the revised version of our paper.
>
> **Dataset Filtering**
>
> Weak Model Filtering: We use DeepSeek-R1-Distilled-Qwen-7B to perform 16 rounds of filtering on math problems, removing those with a pass rate greater than 90%.
>
> Strong Model Filtering: We use DeepSeek-R1-0528 models to perform one round of filtering on math problems, collecting those that cannot be solved.
>
> Problem Rewriting: For problems that failed the strong model filtering step, we suspect that some may have incorrect formats or may be proof-based. Therefore, we use SOTA models (such as DeepSeek-V3) to attempt to rewrite them into problems that can be programmatically verified.
>
> Refiltering: For the problems rewritten in the rewriting step, we reapply both the weak model filtering and strong model filtering, and remove those problems.
>
> **Verifier Configuration**
>
> Output Format: We prompt the model to think step by step and place the output inside \boxed{}, using the following prompt template (in Python f-string form):
>
> ```
> f"{problem}\nPlease reason step by step, and put your final answer within \\boxed{}."
> ```
>
> Verification: We use the open-source Math-Verify library (https://github.com/huggingface/Math-Verify) to extract the last parseable mathematical expression from the model's final answer and compare it to the reference solution using the Math-Verify library.
> This verifier configuration can be used to validate both purely numeric answers and expressions that contain variables.
>
> # 10. [W8] Hyperparameter tables, seeds, and scripts should be released for full reproducibility.
>
> Thank you for your suggestion.
>
> For the hyperparameter tables, we will add this information in appendix later for easy reference.
>
> For the training scripts, we will release them publicly after the paper is officially accepted.

---

> ### Author Response · Authors · 2025-11-27
> **Hoping for Your Reassessment!**
>
> We sincerely thank the reviewers for their insightful comments. We have addressed all the concerns raised in our responses and hope that our clarifications enhance both the clarity and understanding of our work. We plan to incorporate the relevant content into the revised manuscript and hope that the reviewers will consider reevaluating our work.

---

### Official Review · Reviewer_c84Y · 2025-11-03

**Soundness:** 2
**Presentation:** 2
**Contribution:** 2
**Rating:** 2
**Confidence:** 3

**Summary:**

This paper introduces the instability issue of routing mechanism in the reinforcement learning of MoE models, and proposes a post-training method named rollout routing replay (R3) by replaying the inference routing weights when tuning model parameters. Some experimental results verify the performance of R3 on stabilizing RL in MoE models. Furthermore, R3 can be applied with other RL methods, such as GSPO and TIS simultaneously.

**Strengths:**

1. The training and inference discrepancy is shown under different ways.
2. The idea seems to work with different RL algorithms.
3. The paper is well written and the method is clearly explained.

**Weaknesses:**

Justification:
Soundness: Though this paper shows the policy discrepancy between training and inference in multiple ways, such as KL Divergence and Extreme Token Distribution Function introduced in the paper, the reasons behind this phenomenon are not thoroughly studied yet. The experiments are not detailed enough. The increase of computation complexity of R3 is not shown. So, the results of this paper are not very convincing.

Presentation: The figure 1 of this paper is clear to show the algorithm flow. But the meaning of the x-axis in figure 3 is not explained clearly. The training dynamics in figure 6 to show the performance of the algorithm is not explained well. Are there some reference papers comparing in such ways? Another question, what does the notation r(·) in line 157 mean? It does not appear in equation (2).

Contribution: The instability of RL in posting training of MoE is an important problem which is worth further research. This paper proposes to alignment the router weights of samples in training and inference to stable the RL training. The idea is easy to understand and seems to work under the experiment setting. However, the motivation of this idea is not very clear, and the experiment results are not very convincing.

In summary:
1. The motivation of the method is not clear.
2. There are some mistakes in equations and some figures are not clearly shown.
3. The experiments are insufficient. The increase of computation complexity is not shown and the comparison with more SOTA algorithms is missing.

**Questions:**

1. Can you compare the complexity of the proposed method with others?
2. Can you clearly show the relation between the training-inference discrepancy and training instability?
3. Can you add more experiment results of your method compared to other SOTA algorithms?

---

> ### Author Response · Authors · 2025-11-27
> **A clear explanation of the motivation behind our method (Part 1)**
>
> We sincerely appreciate the time and effort you have devoted to reviewing our manuscript. Your insightful comments have prompted us to reflect deeply on the logical flow of our work. We have noted your concern that the motivation behind our method may not be sufficiently clear and that the underlying causes of training-inference discrepancy were not thoroughly investigated.
>
> We respectfully believe there may be some misunderstanding regarding our work, which could stem from a lack of familiarity with the nondeterminism of floating-point arithmetic in machine learning systems-a knowledge gap that is indeed common among some AI researchers.
>
> We attempted to read our manuscript from the perspective of someone without this background and realized that our paper indeed lacks explicit exposition of this foundational knowledge. We sincerely apologize for this and respectfully ask for the opportunity to clarify our motivation.
>
> In fact, the nondeterminism in machine learning systems and the discrepancies across different frameworks is a basic and complex problem and have already attracted research attention. A well-known blog[1] by the Thinking Machine Lab team investigates several commonly observed cause, such as the non-associativity of floating-point operations and batch invariance.
>
> Below, we revisit and structure these concepts more clearly. We will begin by explaining the non-associativity of floating-point arithmetic, then discuss how this phenomenon affects modern models and what additional challenges emerge in MoE architectures. With this foundation, the motivation behind our work becomes clear in a natural progression.
>
> ---
>
> # Fundamental reason: Floating-point addition is non-associative
>
> It is well understood that floating-point addition is non-associative, meaning that the final numerical result may vary depending on the order of operations. The reason is that floating-point numbers have limited precision, and every arithmetic step introduces rounding. For instance, evaluating 1e20 + (-1e20) + 1 yields a different result than 1e20 + 1 + (-1e20). This can be easily reproduced using Python. The first one will yield 1 and the second one will yield 0. The reason is that 1e20 + 1 is rounded to 1e20 for the second expression, so the intermediate result loses the 1. This behavior becomes especially noticeable in iterative computations or parallel reductions, where the execution order is not guaranteed. The non-associativity of floating-point addition leads to several higher-level sources of uncertainty, which are: Floating-Point Atomic Operations, No Batch Invariance, and Variation in Operator and Kernel Selection.
>
> ## Floating-Point Atomic Operations
>
> When atomic addition is applied to floating-point values, the execution order of concurrent updates cannot be strictly guaranteed, leading to non-deterministic and unreproducible outcomes. A representative case is the scatter_add_ operator in PyTorch, which allows repeated indices in the "index" tensor. When multiple values are accumulated into the same memory address, the summation order depends on the hardware scheduling behavior rather than a fixed rule, and therefore the output cannot be guaranteed to be identical across runs.
>
> ## No Batch Invariance
>
> Another source of discrepancy arises when the model processing pipeline does not satisfy batch invariance. In inference framework, new sequences continuously enter the batch while completed sequences exit. Because the batch composition changes dynamically, an identical input sequence may occupy different positions in the batch across multiple forward passes. The blog [1] has pointed out that this dynamic reordering may lead to discrepancies in output probability distributions.
>
> This phenomenon may also affect the training framework. For instance, when computing the outputs of an old policy and a new policy on the same dataset--common in reinforcement learning or PPO-style optimization--differences in sample order within a batch may yield statistically significant deviations, even though the model parameters are identical.
>
> (See continuation below)
>
> ---
>
> Reference:
>
> [1]. He, Horace and Thinking Machines Lab, "Defeating Nondeterminism in LLM Inference", Thinking Machines Lab: Connectionism, Sep 2025. URL: https://thinkingmachines.ai/blog/defeating-nondeterminism-in-llm-inference/

---

> ### Author Response · Authors · 2025-11-27
> **A clear explanation of the motivation behind our method (Part 2)**
>
> ## Variation in Operator and Kernel Selection
>
> Variation in operator and kernel selection can lead to numerical differences between training and inference framework. Differences in operator implementations across frameworks may change the reduction order in floating-point operations, resulting in small deviations. Additionally, even when high-level frameworks such as PyTorch appear to use the same operators, different batch sizes may trigger different backend kernels. For example, matrix multiplication operations on gpu may select various backend kernels depending on matrix size, data type, batch configuration, and hardware architecture to optimize performance. While these kernels are mathematically equivalent, dynamic scheduling can modify the sequence of floating-point accumulation. Consequently, different runtime conditions or batch settings may introduce slight numerical divergence.
>
> # Discrepancy at the Model Level
>
> ## Small Numerical Diff Can Build Up Across Layers
>
> Large language models stack many layers. Tiny numerical differences--for example from different floating-point reduction orders--can enter at one layer and then pass to the next. Because each layer transforms its input, these small changes can slowly grow and change the model’s internal activations. Over many layers, this can shift the final probability outputs between runs or environments.
>
> ## MoE Models Make This Worse: the Router is Discontinuous
>
> MoE models add a new source of instability. The router uses a top-K selection, which is not a smooth function: tiny changes in routing scores can flip which experts are picked. When routing changes, the model follows completely different computation paths. That discrete switch can greatly magnify small numeric differences, producing much larger end-to-end behavior changes. This is particularly harmful for reinforcement learning, where different action distributions or rollout behaviors can hurt training.
>
> ## Full, Bit-for-Bit Alignment is Very Costly
>
> One way to avoid these differences is to force full floating-point alignment so every reduction and operator behaves identically. In practice that means changing and tightly aligning many low-level operators and the order of reductions to guarantee batch invariance. Doing this is expensive: it needs a lot of engineering work, careful operator edits, and it can slow down runtime noticeably. (The implementation in [1] has a 34.35% overhead, and the implementation in [2] has a 61.5% overhead, while our R3 has only a 3.45% overhead.)
>
> ## R3 is a Practical Fix for MoE Router Discrepancy
>
> At this point, we can naturally and clearly state the design motivation of R3. R3 targets the router as the main amplifier of discrepancy. Instead of trying to make every floating-point operation identical, R3 replays inference routing to keep router choices consistent across runs. With only a small extra time cost, R3 significantly reduces the training–inference Discrepancy in MoE models. That lower discrepancy helps stabilize behavior and is beneficial for reinforcement learning training.
>
> ---
>
> We will include these discussion in the revised manuscript, specifically in Section 2 and the Appendix. We hope that these additions can help you to revisit and more accurately assess our work. In the following sections of this response, we will begin addressing the weaknesses and questiones you raised.
>
> ---
>
> Reference:
>
> [1]. SGLang Team. 2025. Towards Deterministic Inference in SGLang and Reproducible RL Training. LMSYS Org Blog, Sep, 2025. URL: https://lmsys.org/blog/2025-09-22-sglang-deterministic/
>
> [2]. He, Horace and Thinking Machines Lab, "Defeating Nondeterminism in LLM Inference", Thinking Machines Lab: Connectionism, Sep 2025. URL: https://thinkingmachines.ai/blog/defeating-nondeterminism-in-llm-inference/

---

> ### Author Response · Authors · 2025-11-27
> **Response to your questions and weaknesses (part 3)**
>
> # 1. [Q1] Can you compare the complexity of the proposed method with others?
>
> Thank you for the question. We compare three categories of approaches: IS-correction-style methods(like GSPO and TIS), R3, and fully aligned true on-policy training and inference frameworks. From a theoretical perspective, all methods share the same computational complexity of **O(T)**, where T denotes the number of tokens.
>
> To more accurately quantify practical differences, we report the additional computational overhead relative to the baseline method and show the results as percentage increase in runtime. The results are shown below.
>
> | Method | Additional Time Cost(compared with the baseline) |
> |--------|-----------------------|
> | Baseline (GRPO) | 0 |
> | IS-correction-style methods (GSPO, TIS, etc.) | almost 0 |
> | R3 | 3.45% |
> | Reproducible RL[1] | 34.35%|
> | True on-policy[2] | 61.5%|
>
> For R3, we conduct end-to-end performance benchmarking to provide a clear comparison. Specifically, we selected the Qwen-30B-A3B model and performed reinforcement learning training under two settings: with R3 and without R3. The experimental configuration is as follows: batch_size = 256, N=8, max_length = 32K.
>
> The measured runtime represents true end-to-end execution latency, including inference, forward passes, backward passes, model optimization, and communication overhead between inference and training framework. The final benchmarking results are summarized in the table below.
>
> | Method | # Token | Runtime(second) | Runtime(hour) | # Token/s | Relative time cost (compared with the baseline) |
> |--------|---------|---------|---------|---------|-----------------------|
> | w/o R3 | 203989395 | 4820 | 1.339 | 42321 | 100% |
> | w/ R3 | 203201992 | 4967 | 1.380 | 40910 | 103.45% |
>
> Our experiments indicate that, after careful optimization, the inclusion of R3 results in only a 3.45% reduction in end-to-end training throughput. This overhead is minimal and suggests that R3 can be integrated into real-world systems with negligible performance impact.
>
> In summary, our method introduces only approximately 3.45% additional overhead, and we believe that achieving stable training at such a minimal cost is a highly worthwhile selection.
>
>
> # 2. [Q2] Can you clearly show the relation between the training-inference discrepancy and training instability?
>
> Thank you for raising this point. The relationship between the training-inference discrepancy and training instability is already empirically investigated and demonstrated in our manuscript. (see Sec. 5.2, line 415 and Fig. 5 and 6). We summarize more clearly here:
>
> We observed that training runs without R3 show a rapidly increasing training-inference KL divergence and extreme-token statistics, and these increases were ususally associated with gradient norm explosion, reward collapse, abnormal sequence lengths (either excessively short or long), and a rapid decline in evaluation performance. In contrast, runs with R3 maintained KL and extreme-token statistics at very low levels throughout training, and training processes also remain stable.
>
> This evidence demonstrates that the discrepancy between the training and inference distributions is a key factor driving instability, and that reducing this discrepancy leads to significantly more stable reinforcement learning.
>
> # 3. [Q3] Can you add more experiment results of your method compared to other SOTA algorithms?
>
> Thank you for the suggestion. Our current experiments already include strong and widely recognized baselines, such as **GSPO**[3] and **TIS**[4]. These methods represent the main categories in this area (TIS for reweight IS, and GSPO for reweight IS and clip), and the comparison results show that R3 achieves clear and consistent improvements.
>
> We would also like to emphasize that R3 is **orthogonal** to existing approaches, meaning it can be combined with methods like GSPO and TIS rather than replacing them. Because of this relationship, the goal of our evaluation is not to compare against every possible method, but to show the performance that R3 brings as an independent module.
>
> Based on the above, we believe the current comparison already provides a fair and representative evaluation of the proposed method, which is sufficient to support the main conclusions of this paper.
>
> ---
>
> Reference:
>
> [1]. SGLang Team. (2025). Towards Deterministic Inference in SGLang and Reproducible RL Training. LMSYS Org Blog. https://lmsys.org/blog/2025-09-22-sglang-deterministic/
>
> [2]. He, Horace and Thinking Machines Lab. (2025). Defeating Nondeterminism in LLM Inference. Thinking Machines Blog. https://thinkingmachines.ai/blog/defeating-nondeterminism-in-llm-inference/
>
> [3]. Zheng, C., et al. (2025). Group sequence policy optimization. arXiv. https://arxiv.org/abs/2507.18071
>
> [4]. Yao, F., et al. (2025). Your efficient RL framework secretly brings you off-policy RL training. Notion. https://fengyao.notion.site/off-policy-rl

---

> ### Author Response · Authors · 2025-11-27
> **Response to your questions and weaknesses (part 4)**
>
> # 4. The experiments are not detailed enough
>
> Thank you for your feedback. We will add more detailed experiment settings as well as new experiment results in the updated version of the paper.
>
> Regarding the current experiment details, the paper already includes a complete description of the training and evaluation setup, and the appendix contains detailed metric curves. We hope you had the chance to notice these parts.
> Reviewer eo2q and Q997 have already evaluated our experiments as “**well-done in terms of benchmarks and baselines**” and “**convincing**”.
>
> We will also continue to expand the descriptions, including dataset filtering steps, validator settings, and a complete list of hyperparameters in our revised paper.
>
> In response to requests from other reviewers, we have also added several new experiment results, including:
>
> ### Results on the new model: DeepSeek-V2-Lite
>
> Setting: Mini Step=1, Model=DeepSeek-V2-Lite(SFT on Mixtrue-of-Thought dataset), Learning Rate=3e-6
>
> | Method   | AIME24      | AIME25      | AMC         | Math500(lv5)| Avg Val Score   | Crash Step |
> |--------  |-------------|-------------|-------------|-------------|-------------    |------------|
> | w/o R3   | 6.87(60)    |  9.69(55)   | 42.50(50)   | 37.50(55)   | 23.55(55)       | 100        |
> | w/ R3    | 8.85(210)   | 11.46(145)  | 48.28(165)  | 43.28(235)  | 26.43(235)      | -          |
>
>
> ### Results on the new task: SWE-bench-Verified
>
> Setting: Mini Step=1, Model=Qwen3-30B-A3B, Learning Rate=3e-6
>
> | Method    | SWE-bench-Verified        | Best Global Step | Crash Step |
> |-----------|---------------------------|------------------|------------|
> | w/o R3    | 31.80                     | 70               | 90         |
> | w/ R3     | 38.60                     | 160              | -          |
>
>
> #### Results of multiple runs under different random seeds for GSPO, TIS, and R3
>
> Setting: Mini Step=4, Model=Qwen3-30B-A3B-Base, Learning Rate=2e-6
>
> | Method | seed | AIME24      | AIME25      | AMC         | Math500(lv5)| Val Score   |
> |--------|------|-------------|-------------|-------------|-------------|-------------|
> | GSPO   | 2025 | 56.35       | 36.56       | 90.47       | 87.13       | 67.63       |
> | GSPO   | 2026 | 45.21       | 35.52       | 85.00       | 80.04       | 61.44       |
> | GSPO   | 2027 | 55.73       | 34.38       | 90.31       | 87.50       | 66.98       |
> | TIS    | 2025 | 47.81       | 33.13       | 86.41       | 84.89       | 63.06       |
> | TIS    | 2026 | 49.58       | 32.50       | 86.56       | 84.33       | 63.24       |
> | TIS    | 2027 | 51.56       | 35.00       | 88.28       | 85.82       | 65.16       |
> | R3     | 2025 | 57.54       | 39.06       | 91.41       | 88.81       | 69.20       |
> | R3     | 2026 | 56.04       | 35.31       | 89.06       | 88.43       | 67.21       |
> | R3     | 2027 | 52.81       | 37.92       | 88.13       | 84.14       | 65.75       |
>
> | Method | Mean±Std    |
> |--------|-------------|
> | GSPO   | 65.35±3.40  |
> | TIS    | 63.82±1.16  |
> | R3     | 67.39±1.73  |
>
> These new results, together with the added details, will be included in the revised manuscript. We expect that these additions will make our experimental section more complete and better demonstrate the effectiveness of R3, as well as its applicability across different settings and tasks.
>
> # 5. The meaning of the x-axis in figure 3 is not explained clearly.
>
> Thank you for your suggestion. Below, we formally define the meanings of the x-axis and y-axis in Figure 3 using mathematical notation.
>
> Let $D = \{s_1, s_2, \ldots, s_B\}$ represent a batch of sequences.
> For each sequence $s_i$, let its length be $N_i$, and let the model depth be $L$.
> Define the router difference at forward pass as:
>
> $$
> d(i, t, l) = \text{difference between router selection during training and inference at router of layer l at token }s_{i, t}
> $$
>
> **Figure 3(a)**
>
> $$
> y = \frac{1}{L \cdot \sum_{i=1}^{B}N_i}
> \sum_{i=1}^{B} \sum_{t=1}^{N_i} \sum_{l=1}^{L}
> \mathbf{1}[d(i,t,l)=x]
> $$
>
> **Figure 3(b)**
>
> $$
> y = \frac{1}{\sum_{i=1}^{B}N_i}
> \sum_{i=1}^{B} \sum_{t=1}^{N_i}
> \mathbf{1}\left[\sum_{l=1}^{L} d(i,t,l)=x\right]
> $$
>
> **Figure 3(c)**
>
> $$
> y = \frac{1}{B}
> \sum_{i=1}^{B}
> \mathbf{1}\left[ x \le \frac{1}{N_i}
> \sum_{t=1}^{N_i}\sum_{l=1}^{L} d(i,t,l) < x+1 \right]
> $$
>
> These definitions will be included in the appendix for reference.

---

> ### Author Response · Authors · 2025-11-27
> **Response to your questions and weaknesses (part 5)**
>
> # 6. The training dynamics in figure 6 to show the performance of the algorithm is not explained well. Are there some reference papers comparing in such ways?
>
> Yes! In the area of reinforcement learning for large language models, it is very common to track **test set accuracy, response length, entropy, and gradient norm** when studying training dynamics. For researchers who run experiments frequently, the importance of these metrics is obvious and widely recognized.
>
> Well-known works such as **DAPO[1]**, **VAPO[2]**, **Cov-Clip[3]**, and **20/80 entropy rule[4]** analyze response length, entropy and test accuracy when discussing training dynamics. The work **TIS[5]** which studies the training-inference discrepancy also reports response length, entropy, and test accuracy during training.
>
> The gradient norm is also especially important because it reflects optimization stability. This is true for almost any training process, including pre-training, SFT, and reinforcement learning.
>
> For these reasons, the metrics used in Figure 6 follow common practice in the field and are appropriate for evaluating RL training dynamics for LLM.
>
> # 7. What does the notation r(·) in line 157 mean? It does not appear in equation (2)
>
> Thank you for pointing out this mistake and we sincerely apologize for the oversight.
>
> The reference to r(·) has been removed, as it was mistakenly included. In addition, Equation (2) also contained an error: the denominator should be $\pi_{\text{infer}}$ instead of $\pi_{\text{train}}$. We have corrected this issue, and the updated equation (2) is now:
>
> $$
> D_{\text{KL}}(\pi_{\text{train}}(\theta)|| \pi_{\text{infer}}(\theta))
> \approx \frac{1}{|T|}\sum_{t \in T}\bigl[\frac{\pi_{\text{train}}(\theta)(t)}{\pi_{\text{infer}}(\theta)(t)}-1 - \log \frac{\pi_{\text{train}}(\theta)(t)}{\pi_{\text{infer}}(\theta)(t)}\bigr].
> $$
>
> ---
>
> Reference:
>
> [1]. Yu, Q., et al. (2025). DAPO: An open-source LLM reinforcement learning system at scale. arXiv. https://arxiv.org/abs/2503.14476
>
> [2]. Yue, Y., et al. (2025). VAPO: Efficient and reliable reinforcement learning for advanced reasoning tasks. arXiv. https://arxiv.org/abs/2504.05118
>
> [3]. Cui, G., et al. (2025). The entropy mechanism of reinforcement learning for reasoning language models. arXiv. https://arxiv.org/abs/2505.22617
>
> [4]. Wang, S., et al. (2025). Beyond the 80/20 rule: High-entropy minority tokens drive effective reinforcement learning for LLM reasoning. arXiv. https://arxiv.org/abs/2506.01939
>
> [5]. Yao, F., et al. (2025). Your efficient RL framework secretly brings you off-policy RL training. Notion. https://fengyao.notion.site/off-policy-rl

---

> ### Author Response · Authors · 2025-11-27
> **Request for Your Reconsideration**
>
> In summary, we have thoroughly clarified the motivation behind our approach and addressed all of your questions. We sincerely hope that, in light of our responses, you will reconsider your evaluation of our work. We would be deeply grateful for your thoughtful reassessment.

---

### Official Review · Reviewer_eo2q · 2025-11-03

**Soundness:** 4
**Presentation:** 4
**Contribution:** 3
**Rating:** 6
**Confidence:** 3

**Summary:**

This paper studies the issue of RL stability in LLM fine-tuning from the perspective of output distribution divergence between inference and training, an inevitability due to kernel, floating point operation, and computation graph non-determinism. The authors identify MOE routing non-determinism as a key contributor to high KL-divergence between sampler and trainer probability distributions, and propose a simple method for aligning routing by caching the routing masks computed at inference time and replaying them at training time. Experimentally, the method results in more stable and effective RL training on Qwen3-3b.

**Strengths:**

Improving the stability of RL training for LLMs is a very important yet understudied topic. This paper clearly identifies a root issue that is relevant for most modern models (MoE routing), and provide multiple illustrations of the significance of the issue in section 3. Furthermore, the solution (R3) is simple and easy-to-understand. Experimentally, the proposed method seems to work better than the standard fix of importance sampling, which is known to also introduce stability issues. The experiments are well-done in terms of benchmarks and baselines.

**Weaknesses:**

The main issue is that the proposed fix is only evaluated on Qwen3-30b, and not at larger or smaller model scales or different MoE LLMs. For example, Section 3 suggests that Qwen3-8b does not suffer from this issue, so it's hard to evaluate how broadly applicable this fix beyond this exact model. Therefore, I'm willing to raise my score if a more comprehensive evaluation is provided.

**Questions:**

1. Is gating noise applied to the router during training? If so, what is the schedule of the noise, and the potential impact on trainer-inference divergence?

---

> ### Author Response · Authors · 2025-11-28
> **Response to your questions and weaknesses**
>
> We truly appreciate the time and effort you put into reviewing our submission. Your feedback and analysis not only highlight the strengths of our work but also help us identify areas for improvement. Thank you for recognizing the importance of training-inference discrepancy and for understanding the motivation behind our work, as well as for your positive assessment of our benchmarks and baselines in our experiments. In response to your questions and concerns, we provide the following clarifications and additional results:
>
> # Lack of performance validation on other different models.
>
> **R3 also demonstrates superior performance on DeepSeek-V2-Lite**
>
> Thank you for your suggestion! To further validate the generality of the proposed R3 method, we additionally conducted experiments on the DeepSeek-V2-Lite model, which adopts a shared-expert architecture with 2 fixed shared experts and 6 experts are dynamically selected.
> We first SFT it using the open-source reasoning dataset Mixture-of-Thought [1], resulting in a strengthened version referred to as DeepSeek-V2-Lite-Reasoning. We then performed reinforcement learning training on this model without R3 and with R3. The results are summarized in the table below.
>
> (The numbers outside the parentheses indicate the best metrics across all checkpoints, while the numbers inside the parentheses indicate the corresponding training steps.)
>
> | Method   | AIME24      | AIME25      | AMC         | Math500(lv5)| Avg Val Score   | Crash Step |
> |--------  |-------------|-------------|-------------|-------------|-------------    |------------|
> | w/o R3   | 6.87(60)    |  9.69(55)   | 42.50(50)   | 37.50(55)   | 23.55(55)       | 100        |
> | w/ R3    | 8.85(210)   | 11.46(145)  | 48.28(165)  | 43.28(235)  | 26.43(235)      | -          |
>
> The experimental results demonstrate that the R3 method substantially improves training stability. Without R3, the training collapses at about 100 steps. With R3, the training remains stable and can continue up to 250 steps without any signs of instability.
>
> We hope that the results obtained on the new model help alleviate your concerns regarding the generalizability of our approach. If you require more additional experiments, please let us know, and we would be glad to provide further experimental results.
>
> # Why Qwen3-8b does not suffer from this issue?
>
> **Qwen3-8B is a dense model, not MoE model.**
>
> Thank you for your question! As stated in Section 3 of the original manuscript, our work specifically focuses on MoE architectures, aiming to address the additional instability observed in MoE models compared to dense models. We attribute this additional instability primarily to the routing behavior introduced by the MoE router mechanism. The proposed R3 method is explicitly designed for MoE architectures, with the goal of improving their stability to a level comparable to dense models.Since Qwen3-8B is a dense model without a router component, it is inherently unaffected by the instability issues that R3 aims to resolve.
>
> # Is gating noise applied to the router during training?
>
> **No**
>
> We check our implementation carefully. We confirm the following:
>
> a. No gating noise or load-balancing strategy was applied to the MoE router during reinforcement learning training;
>
> b. The router parameters were updated solely through gradient backpropagation and the optimizer

---

> ### Author Response · Authors · 2025-11-28
> **Looking Forward to Your Feedback!**
>
> We hope that the results obtained on the new model help alleviate your concerns regarding the generalizability of our approach. If you require more additional experiments, about new models or gating noise, please let us know, and we would be glad to provide further experimental results.

---

### Author Response · Authors · 2025-12-04

We thank the reviewers for their careful reading and for the constructive comments and suggestions.

We have revised the manuscript to address the reviewers’ weaknesses and questions. Below we summarize the main changes.

Main text

* Section 2: We added a background discussion on nondeterminism in modern LLM training and inference systems to help readers who are not familiar with this background better understand the motivation of our paper. (for c84Y)
* Section 5.3: We added performance test results in Section 5.3. (for c84Y, SRHz)
* Section 6: We shortened the related work in Section 6 and added discussion on nondeterminism, framework discrepancies, and their effects on reinforcement learning. (for SRHz, Q997)
* Section 7: We added a discussion section in Section 7 explaining why R3 is more effective than loss correction(GSPO, TIS), in particular, R3 reduces gradient noise and stabilizes importance sampling. We also outlined future work directions in two streams: infrastructure-level work and algorithm-level work. (for Q997)

Appendix

We added many new experiments, details, and analyses in the appendix:

* Appendix B.1: Experiments on DeepSeek-V2-Lite. (for eo2q)
* Appendix B.2: Experiments on Qwen3-8B. (for eo2q)
* Appendix B.3: Validation of R3’s effect on training-inference consistency across multiple MoE models. (for eo2q)
* Appendix C: Experiments with fixed router. (for Q997)
* Appendix D: Multi-turn reinforcement learning experiments on the SWE task. (for SRHz)
* Appendix E: Random seed experiments. (for SRHz)
* Appendix F: End-to-end performance test detail. (for SRHz, c84Y)
* Appendix G: Memory usage analysis. (for SRHz)
* Appendix H: Full definitions of the metrics in Section 3.2. (for c84Y)
* Appendix I: Experiments on explicit reasoning models.
* Appendix J: Dataset filtering details. (for SRHz)
* Appendix K: Details of the verifier configuration. (for SRHz)
* Appendix L: Analysis of token-type specific routing discrepancies. (for Q997)
* Appendix M: List of hyperparameters. (for SRHz)

We believe these revisions address the reviewers’ concerns and make the manuscript clearer and more complete.

---

### Author Response · Authors · 2025-12-04
**Summary for AC (Part 1)**

Dear Area Chair,

Thank you very much for taking the time to read our paper. Below we give a short summary to help you quickly get into our paper and the reviewers' comments.

---

### Paper TL;DR:
We found that discrepancies between the training framework and the inference framework cause different MoE router choices. This makes the target policy and the sampling policy in RL very different. This leads to instability in MoE models during RL. We propose Rollout Routing Replay(R3) to fix this problem by replaying routing distribution.

### Reviewers noted these strengths of our paper:

1. The problem is important

   eo2q: "Improving the stability of RL training for LLMs is a very important yet understudied topic."

   SRHz: "Stabilizing MoE RL is a highly relevant and urgent problem in LLM post-training."

2. Strong motivation

   Q997: "The paper identifies precisely the core problem: routing inconsistencies in MoE models during RL training."

   Q997: "This systematic breakdown provides a strong foundation for the proposed solution."

   SRHz: "The paper isolates a concrete but under-explored source of RL collapse: router nondeterminism in MoE models."

   eo2q: "This paper clearly identifies a root issue that is relevant for most modern models (MoE routing), and provide multiple illustrations of the significance of the issue in section 3."

3. Method is simple and clear

   Q997: "R3 is a simple yet effective mechanism.

   SRHz: "The proposed fix, replaying inference routing distributions into training, is conceptually simple yet novel"

   eo2q: "Furthermore, the solution (R3) is simple and easy-to-understand."

4. Convincing experiments

   Q997: "The experimental results are convincing"

   SRHz: "The empirical evaluation demonstrates evidence of reduced policy KL and improved stability."

   eo2q: "The experiments are well-done in terms of benchmarks and baselines."

### Main concerns raised by reviewers:

* More experiments (more models, more tasks, random seed experiment). (eo2q, SRHz)

* Speed and memory cost. (SRHz)

* Lack of background on training vs inference discrepancies. (c84Y, Q997)

* More details and analysis (more experiment details, routing discrepancies for certain token types). (SRHz, Q997)

* some other smaller, individual questions.

We answered all these points in detail. We believe our revision fixes all reviewer concerns and makes the paper clearer and more complete.

### After revision, our paper now has:

1. An important research question.

2. Very strong motivation.

3. A method that is simple and easy to understand.

4. A full and detailed experimental study, including:

* Different models (Qwen, Deepseek-V2-Lite, Mixtral, Moonlight)

* Different model stages (Base model, Instruct SFT model, Reasoning SFT model)

* Different tasks (Math, SWE-Bench)

* Multiple random seeds to ensure confidence

5. Full analysis and discussion, including:

* Analysis of gradients, entropy, and generation length

6. High practical feasibility, including:

* Throughput overhead: only 3.45% overhead

* Memory overhead: only +1.5 GB CPU memory per machine

Specifically, the random seed experiment and the SWE task experiment cost us a lot of resources, time, and effort. The random seed experiment used many GPU hours. The SWE task required building a complex K8s cluster and an agent interaction system. We think this work was worth it because these experiments confirm result confidence and show R3 works in popular agentic tasks.

---

> ### Author Response · Authors · 2025-12-04
> **Summary for AC (Part 2)**
>
> ### Concern about reviewer c84Y:
>
> We found reviewer c84Y gave a low score (Rating 2). They said our motivation was weak and the experiments were not convincing. We were very surprised and take this comment seriously.
>
> We read that review carefully and think the reviewer misunderstood our paper. We think the reasons may include:
> * The reviewer is not familiar with the discrepancies between training and inference frameworks. They might think the discrepancies is only due to different math algorithm.
> * The reviewer is not familiar with RLVR-related work for LLMs. The reviewer said our dynamic analysis is weak, but such metrics are normal in RLVR work.
>
> We prepared a long reply to help. We explain the basic facts about differences between training and inference frameworks and give an intro to RLVR work. We hoped this would help the reviewer understand our motivation. Sadly, on November 28 the reviewers are not allowed to discuss. This blocked our chance to discuss with the reviewer. We truly hope the Area Chair can help clarify this misunderstanding, so that the paper may receive a fair and balanced evaluation. We are sincerely grateful for your time and support.
>
>
> ### Progress in the open source community:
>
> **Anonymity note: the paper authors did not take part in these projects, and the links do not reveal the authors' identity. We list them only to show that our method is important.**
>
> 1. On October 27, the open source project SGLang added support to return router distribution info from an MoE model's inference framework, showing support for Rollout Routing Replay. ([https://github.com/sgl-project/sglang/pull/12162](https://github.com/sgl-project/sglang/pull/12162)).
> 2. On November 13, the open source project slime added support for Rollout Routing Replay. ([https://github.com/THUDM/slime/pull/715](https://github.com/THUDM/slime/pull/715)).
> 3. On December 1 (67 days after the ICLR submission deadline), Deepseek-V3.2 tech report was released ([https://huggingface.co/deepseek-ai/DeepSeek-V3.2/blob/main/assets/paper.pdf](https://huggingface.co/deepseek-ai/DeepSeek-V3.2/blob/main/assets/paper.pdf)). It uses the same method as our paper (see page 8, "Keep Routing" subsection). As far as we know, this is the second public presentation of this method (the first was our paper).
>
> ---
>
> We sincerely hope that you could review this summary and assist the other reviewers in making a fair reassessment of our revised work. Again, thank you for your important efforts in supporting the stability of the ICLR community!
>
> Sincerely
>
> The authors

---

### Meta-Review · Area_Chair_kKwq · 2026-01-05

**Summary:**

The reviewers generally agree that the paper identifies a critical and timely problem: the instability of RL training in MoE architectures caused by non-deterministic routing. The proposed solution, Rollout Routing Replay (R3), is recognized for its simplicity, low computational overhead, and empirical effectiveness on mathematical reasoning benchmarks. However, the initial submission suffered from significant technical and presentation issues that informed the recommendation for rejection. Primary concerns included a lack of generalizability across different model scales and architectures, pervasive notation errors (specifically in the PPO and KL divergence equations), and a lack of clear theoretical or technical depth regarding why the routing discrepancy occurs and why R3 specifically outperforms existing methods like (Total) Importance Sampling. While the problem is high-impact, the consensus suggests the paper requires a more rigorous foundation and polished presentation to meet the standards for acceptance.

**Reviewer Concerns:**

The authors provided an exceptionally thorough rebuttal that addressed many of the reviewers' primary concerns. Specifically, they successfully addressed the generalization concern by providing new results on DeepSeek-V2-Lite and extending the task domain to SWE-bench-Verified, where R3 showed significant stability gains. The authors also clarified the computational complexity, providing concrete data that R3 introduces only a 3.45% overhead, far lower than full bit-for-bit alignment methods. Furthermore, the detailed explanation of floating-point non-associativity and kernel selection provided a much-needed technical grounding for the "motivation" concerns raised by reviewers. However, some issues remain outstanding. The initial manuscript contained fundamental errors in core equations (e.g., the denominator in the KL divergence formula and the inclusion of non-existent notations like r(\cdot)). While corrected in the rebuttal, these errors suggest a lack of initial rigor. Additionally, the critique remains that R3 is more of a practical engineering fix for framework non-determinism rather than a fundamental algorithmic breakthrough in reinforcement learning theory.

**Reviewer Scores:**

Reviewer eo2q: Originally a 6. Given the addition of DeepSeek-V2-Lite results and the clarification on dense vs. MoE behavior, this reviewer would likely have stayed at a 6 or moved to a 7, as their main "weakness" regarding comprehensive evaluation was directly addressed.

Reviewer Q997: Originally a 6. The detailed token-level analysis showing that mathematical tokens suffer the most from discrepancy likely would have satisfied this reviewer’s curiosity. They would likely maintain a 6 or 7.

Reviewer SRHz: Originally a 4. The rebuttal addressed almost all of this reviewer’s technical questions (GPU-hour quantification, multi-seed results, and equation fixes). They would likely have moved to a 6.

Reviewer c84Y: Originally a 2. Although the authors provided a very detailed tutorial on floating-point non-determinism to clarify the motivation, this reviewer’s skepticism regarding the "contribution" being "fair" suggests they view the work as too incremental. They might have moved to a 3 or 4, but likely remained a "Reject" vote based on the simplicity of the replay mechanism.

---

### Decision · Program_Chairs · 2026-01-26

Reject